# Adult re-expression of IRSp53 rescues NMDA receptor function and social behavior in IRSp53-mutant mice

Young Woo Noh[1,5], Chaehyun Yook[2,5], Jaeseung Kang[2,5], Soowon Lee[3], Yeonghyeon Kim [1], Esther Yang[4], Hyun Kim[4] & Eunjoon Kim [1,2✉]

IRSp53 (or BAIAP2) is an abundant excitatory postsynaptic scaffolding/adaptor protein that is involved in actin regulation and has been implicated in autism spectrum disorders, schizophrenia, and attention-deficit/hyperactivity disorder. IRSp53 deletion in mice leads to enhanced NMDA receptor (NMDAR) function and social deficits that are responsive to NMDAR inhibition. However, it remains unclear whether IRSp53 re-expression in the adult IRSp53-mutant mouse brain after the completion of brain development could reverse these synaptic and behavioral dysfunctions. Here we employed a brain-blood barrier (BBB)-penetrant adeno-associated virus (AAV) known as PHP.eB to drive adult IRSp53 re-expression in IRSp53-mutant mice. The adult IRSp53 re-expression normalized social deficits without affecting hyperactivity or anxiety-like behavior. In addition, adult IRSp53 re-expression normalized NMDAR-mediated excitatory synaptic transmission in the medial prefrontal cortex. Our results suggest that adult IRSp53 re-expression can normalize synaptic and behavioral deficits in IRSp53-mutant mice and that BBB-penetrant adult gene re-expression has therapeutic potential.

[1] Department of Biological Sciences, Korea Advanced Institute of Science and Technology (KAIST), Daejeon 34141, Korea. [2] Center for Synaptic Brain Dysfunctions, Institute for Basic Science (IBS), Daejeon 34141, Korea. [3] Graduate School of Medical Science and Engineering, KAIST, Daejeon 34141, Korea. [4] Department of Anatomy and BK21 Graduate Program, Biomedical Sciences, College of Medicine, Korea University, Seoul 02841, Korea. [5] These authors contributed equally: Young Woo Noh, Chaehyun Yook, Jaeseung Kang. ✉email: kime@kaist.ac.kr

An emerging topic in psychiatric disorders, including autism spectrum disorders (ASD), schizophrenia, and attention-deficit/hyperactivity disorder (ADHD), is whether certain genes important for brain functions can be restored at adult stages after the completion of brain development to normalize defective brain functions[1,2]. Previous studies on this topic have yielded different answers. For instance, adult re-expression of MeCP2, a Rett syndrome-associated protein, in mice was reported to restore nearly all Rett syndrome-related neurodevelopmental deficits[3–5]. Similarly, adult restoration of GluN1, a subunit of NMDA receptors (NMDARs), in mice was found to rescue most of the GluN1 dysfunction-related phenotypes, including social deficits, repetitive behavior, hyperactivity, and anxiety-like behavior[6]. In contrast, adult restoration of the abundant excitatory postsynaptic protein, Shank3, in mice[7–9] was reported to restore Shank3 dysfunction-related social deficits and repetitive behavior but not hypoactivity or anxiety-like behavior, the latter of which was found to require earlier gene re-expression at embryonic or early postnatal stages[10,11]. Moreover, adult restoration of the abundant excitatory postsynaptic protein, SynGAP1, in mice was shown to rescue SynGAP1 dysfunction-related memory and seizure phenotypes but not hyperactivity or anxiety-like behavior[12–14]. These results suggest that each individual gene associated with brain dysfunction has distinct time windows when certain behavioral deficits may be rescued by gene re-expression, which is an important issue for potential gene-based therapy.

As strategies for gene re-expression, researchers have mainly used genetic approaches, such as by correcting the orientation of flexed target genes. However, adult re-expression of target genes in patient brains would need to involve virus-mediated gene delivery. A recently developed adeno-associated virus (AAV) was designed to penetrate the blood–brain barrier (BBB) by the use of a special capsid protein (PHP.B and PHP.eB) and retro-orbital injection, eliminating the need to use intrathecal AAV injection and maximizing the chance of driving AAV-dependent gene expression in various brain regions and cell types, including neurons and glia[15–17]. However, it remained largely unclear whether the PHP.eB approach could lead to sufficient adult re-expression of brain genes associated with certain brain disorders and substantially rescue related phenotypes in animal models of brain disorders[17–19].

IRSp53 (insulin receptor substrate p53), which is encoded by the Baiap2 gene, is known to promote the polymerization and stability of actin filaments[20,21]. IRSp53 is enriched at excitatory synapses and dendritic spines and promotes the development and maintenance of dendritic spines[20,22–25], which represent actin-filled protrusive neuronal structures that coordinate excitatory synaptic transmission and plasticity[26,27]. Deletion of IRSp53 in mice leads to various synaptic and behavioral abnormalities, including decreased spine density, increased NMDAR function, and cognitive and behavioral impairments[28–32]. Clinically, IRSp53 has been associated with various neuropsychiatric disorders, including ASD[33–35], schizophrenia[36,37], and ADHD[38–42].

IRSp53 expression in rodent brains is detected at low levels at embryonic stages and early postnatal stages, and gradually and steadily increases to adult levels during the first several weeks of postnatal development[23]. This change is consistent with age-dependent increases in the development of actin-rich neuronal processes and dendritic spines[26,27]. However, IRSp53, although expressed at low levels at early stages[23], has been shown to regulate embryonic development and perinatal survival in mice[20]. It is thus unclear whether adult IRSp53 re-expression would be sufficient to rescue the synaptic and behavioral phenotypes in IRSp53-mutant mice, or whether early IRSp53 re-expression during embryonic or early postnatal stages would be required for these phenotypic reversals.

In the present study, we set out to direct adult IRSp53 re-expression using BBB-penetrant IRSp53 expressed from the PHP.eB virus, which was applied via retro-orbital sinus injection. We sought to engineer this gene re-expression in defined neuronal populations by using mice with IRSp53 deletion restricted to excitatory neurons in cortical and hippocampal areas, conditionally driven by the Emx1 promoter (Emx1-Cre mice). This re-expression rescued social and synaptic phenotypes in conditional IRSp53-mutant mice, suggesting that adult IRSp53 re-expression using BBB-penetrating PHP.eB-IRSp53 can rescue IRSp53 dysfunction-related phenotypes in mice.

## Results

### Re-expression of IRSp53 in the adult IRSp53-cKO brain by BBB-penetrant PHP.eB.

A variant of AAV, termed PHP.eB, was previously developed and shown to penetrate and deliver genes to various brain regions at a high efficiency[15]. To test if IRSp53 re-expression in the adult brain could normalize synaptic and behavioral phenotypes, we generated pAAV carrying N-terminally HA-tagged IRSp53 fused to EGFP via the self-cleavable P2A peptide, under control of the neuron-specific human synapsin I promoter (pAAV-hSyn-DIO-mEGFP-GSG-P2A-HA-IRSp53) (Fig. 1a). This construct was used to produce PHP.eB-capsid virus particles (termed PHP.eB-IRSp53 hereafter), which can penetrate the BBB when injected into the retro-orbital sinus (Fig. 1b).

In the mouse brain, PHP.eB-IRSp53 could drive IRSp53/EGFP expression in various Emx1-positive Cre-expressing excitatory neurons in Emx1-Cre;Baiap2[fl/fl] (termed IRSp53-cKO mice hereafter) (Fig. 1c). This was shown by our detection of EGFP fluorescence signals, representing IRSp53 expression, in brain regions including the medial prefrontal cortex (mPFC), anterior cingulate cortex, and hippocampus (Fig. 1d), where the restored levels of IRSp53 proteins were ~60% of wild-type (WT) levels (Fig. 1e and Supplementary Fig. 1). While EGFP signals were mainly detected in cell-body areas, HA signals were mainly detected in neuropil areas (Fig. 1f), suggesting that HA-IRSp53 proteins separated from the fused EGFP by proteolytic cleavage were targeted to dendritic/spine areas. Double fluorescence in situ hybridization for EGFP and Vglut1/2 (glutamate neuron marker) indicated that ~64–86% of EGFP-positive neurons were Vglut1/2 positive (70.5% for prelimbic area, 64.2% for infralimbic area, 85.9% for primary motor area) and that ~55–77% of Vglut1/2-positive neurons were EGFP positive (55.3% for prelimbic area, 62.6% for Infralimbic area, 77.0% for primary motor area) (Fig. 1g–i). These results suggest that PHP.eB-IRSp53 can drive IRSp53 re-expression in Cre-expressing excitatory neurons in various brain regions of adult IRSp53-cKO mice.

### Adult IRSp53 re-expression rescues social interaction and approach in IRSp53-cKO mice.

To test if adult IRSp53 re-expression could rescue social deficits in IRSp53-cKO mice, we injected PHP.eB-IRSp53 or empty PHP.eB-EGFP (control) into the retro-orbital sinus of IRSp53-cKO mice at postnatal week 8 and performed behavioral tests at postnatal week ~12 (Fig. 2a). We also injected PHP.eB-IRSp53 into WT mice to see if IRSp53 overexpression in the WT genetic background could affect social functions.

We found that adult IRSp53 re-expression in IRSp53-cKO mice improved social interaction in the direct social interaction test, where two mice of the same genetic background were allowed to freely interact, as supported by nose-to-nose interaction but not by total body contact time (Fig. 2b). In contrast, adult IRSp53 re-expression in WT mice had no effect on direct social

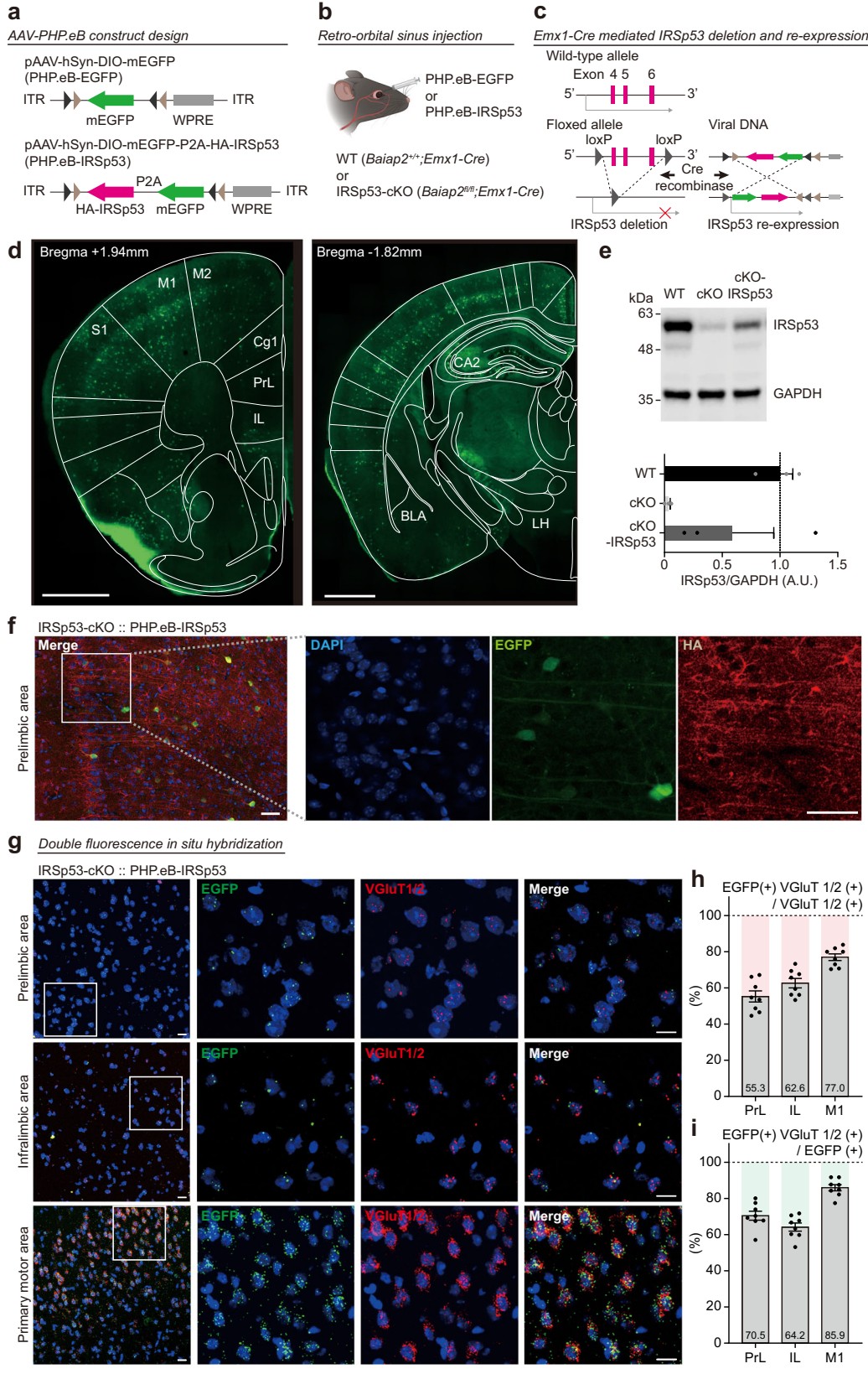

interaction, as judged by nose-to-nose interaction or total body contact time.

IRSp53-cKO mice re-expressing IRSp53 also showed a significantly improved social approach in the three-chamber social-interaction test, as compared with control IRSp53-cKO mice expressing EGFP alone (Fig. 2c, d). In contrast, WT mice expressing additional IRSp53 following PHP.eB-IRSp53 injection showed no alteration of social approach compared to control WT mice expressing EGFP alone. We were unable to determine whether IRSp53 re-expression affects social novelty recognition in IRSp53-cKO mice because WT mice expressing EGFP (positive control) did not prefer a novel stranger over a familiar stranger

**Fig. 1 Re-expression of IRSp53 in the adult IRSp53-cKO brain by BBB-penetrant PHP.eB. a** Diagrams showing the domain structure of the IRSp53 variant used in the present study and the structure of the pAAV variant carrying HA-tagged IRSp53 cDNA (pAAV-hSyn-DIO-mEGFP-GSG-P2A-HA-IRSp53) for the production of BBB-penetrant PHP.eB-IRSp53. Domains of IRSp53: IMD, IRSp53-MIM homology domain; CRIB-PR, CDC42/Rac interactive binding-proline rich domain; SH3, Src homology 3 domain; WW, WW domain; and PDZ-B, PDZ domain-binding motif. pAAV parts; ITR, inverted terminal repeat; hSynI, human synapsin I promotor; mEGFP, monomeric EGFP; HA-IRSp53, N-terminally HA-tagged IRSp53; P2A, a cleavable peptide linker. **b** Diagram showing injection of PHP.eB into the retro-orbital sinus for BBB penetration. **c** Diagram showing a conditional deletion of IRSp53 in Emx1-positive glutamate neurons in *Emx1-Cre;Baiap2/Irsp53^{fl/fl}* (IRSp53-cKO) mice and re-expression of IRSp53 in Emx1-positive glutamate neurons in these mice by the infection of PHP.eB-IRSp53 and Cre-mediated normalization of the orientation of the HA-IRSp53-EGFP cDNA construct. **d** Re-expression of IRSp53 in various brain regions of adult IRSp53-cKO mice, as shown by the presence of EGFP signals, representing IRSp53 expression, in brain regions including the medial prefrontal cortex (mPFC), anterior cingulate cortex (ACC), and hippocampus. Note that the hippocampal CA2 region shows stronger EGFP signals, as previously reported[66]. PHP.eB-IRSp53 was injected into the retro-orbital sinus of IRSp53-cKO mice at postnatal week 8 and allowed to express IRSp53 for 4 weeks. S1, primary somatosensory cortex; M1/2, primary/secondary motor cortex; Cg1, anterior cingulate cortex; PrL, prelimbic region of the mPFC; IL, infralimbic region of the mPFC; CA2, CA2 region of the hippocampus; BLA, basolateral amygdala; LH, lateral hypothalamus. Scale bar, 1 mm. **e** Levels of IRSp53 proteins driven by PHP.eB-IRSp53 are ~60% of WT levels, as shown by immunoblot analysis of IRSp53 proteins in the cortex + hippocampus of WT, IRSp53-cKO, and PHP.eB-IRSp53-infected IRSp53-cKO mice (cKO-IRSp53) (3 months). IRSp53 signals were normalized to GAPDH signals (control) ($n = 3$ mice [WT], 3 [IRSp53-cKO], and 3 [IRSp53-cKO and PHP.eB-IRSp53]). **f** An example showing IRSp53 re-expression in the prelimbic region of the mPFC in IRSp53-cKO mice (3 months), as shown by immunofluorescence staining for EGFP and HA. Note that EGFP signals are mainly found in cell-body areas whereas HA signals are mainly found in neuropil areas, suggesting that HA-IRSp53 proteins separated from the fused EGFP by the proteolytic cleavage of the P2A linker were targeted to dendrite/synaptic areas. DAPI staining was used for nuclear staining. Scale bar, 50 μm. **g** Representative image of the colocalizations between EGFP and Vglut1/2 (glutamatergic neuron marker) in the prelimbic and infralimbic regions of the mPFC and the primary motor area in IRSp53-cKO mice infected with PHP.eB-IRSp53 (8–12 weeks), as shown by double fluorescence in situ hybridization. Scale bar, 20 μm. **h** Quantification of the colocalization between EGFP and Vglut1/2, calculated by EGFP-positive neurons among Vglut1/2-positive neurons ($n = 8$ slices from 2 mice [IRSp53-cKO and PHP.eB-IRSp53]). **i** Quantification of the colocalization between EGFP and Vglut1/2, calculated by Vglut1/2-positive neurons among EGFP-positive neurons ($n = 8$ slices from 2 mice [IRSp53-cKO and PHP.eB-IRSp53]). Error bars represent the standard errors of means.

(Fig. 2e, f). However, the positive social novelty recognition in another IRSp53-cKO mouse line (*CaMKIIα-Cre;Baiap2^{fl/fl}* mice; see below) suggests that the lack of social novelty recognition here may stem from the distinct natures of the two cKO mouse lines (i.e., different cell-type-specific cKOs of the same gene). These results collectively suggest that IRSp53 re-expression in adult IRSp53-cKO mice can improve some aspects of direct social interaction and social approach.

**Adult IRSp53 re-expression does not affect hyperactivity or anxiety-like behavior in IRSp53-cKO mice**. IRSp53-cKO mice were reported to show increased locomotor activity in the open-field test but normal anxiety-like behavior in the elevated plus-maze (EPM)[31]. We thus tested if adult IRSp53 re-expression had any effect on hyperactivity or anxiety-like behavior IRSp53-cKO mice.

EGFP-expressing IRSp53-cKO mice showed greater baseline locomotor activity compared to EGFP-expressing WT mice (Fig. 3a), similar to the hyperactivity observed in naïve IRSp53-cKO mice[31]. Importantly, PHP.eB-IRSp53 injection did not alter locomotor activity in IRSp53-cKO or WT mice, as shown by the lack of a virus effect in two-way ANOVA. In addition, PHP.eB-IRSp53 injection had no effect on the time spent in the center region of the open-field arena by WT and IRSp53-cKO mice. However, EGFP-expressing IRSp53-cKO mice spent less time in the center compared to EGFP-expressing WT mice, which suggested the presence of baseline anxiety-like behavior in the EGFP-expressing IRSp53-cKO mice.

In the EPM test, another anxiety-related test, IRSp53-cKO mice injected with PHP.eB-EGFP (control) did not show altered baseline anxiety-like behavior compared to WT mice, as supported by the lack of a genotype effect (Fig. 3b); this was consistent with a previous report[31]. In addition, PHP.eB-IRSp53 injection did not affect anxiety-like behavior in WT or IRSp53-cKO mice, as shown by the lack of a virus effect.

Intriguingly, in another anxiety-related behavioral test (light-dark test), which was not previously performed for IRSp53-cKO or IRSp53-global-KO mice[29,31], IRSp53-cKO mice injected with PHP.eB-EGFP (control) spent less time in the light chamber and

visited the light chamber less frequently (Fig. 3c), which was suggestive of baseline anxiety-like behavior. However, PHP.eB-IRSp53 injection did not affect these parameters in WT or IRSp53-cKO mice (Fig. 3c).

These results collectively suggest that adult IRSp53 re-expression does not affect the locomotor activity or anxiety-like behavior in WT or IRSp53-cKO mice, although IRSp53-cKO mice showed a moderate increase in baseline anxiety-like behavior. The positive baseline anxiety-like behaviors of IRSp53-cKO mice in the open-field and light-dark tests but not in the EPM test suggest that different anxiogenic stimuli such as openness, height, and brightness may differentially induce anxiety-like behaviors, in line with the distinct natures of these tests[43,44].

Given that the IRSp53 deletion in IRSp53-cKO mice is driven by Emx1-Cre expressed early in embryonic development[45], we attempted late-stage IRSp53 deletion driven by CaMKIIα-Cre known to be expressed at postnatal stages[46]. The results indicated that CaMKIIα-Cre-dependent IRSp53 deletion in mice led to various behavioral deficits, including open-field hyperactivity, anxiety-like behaviors, and impaired social interaction (Supplementary Fig. 2), similar to the behavioral deficits induced by Emx1-Cre-dependent IRSp53 deletion. Therefore, adult-stage IRSp53 expression also seems to be required for normal brain function and behavior in mice.

**Adult IRSp53 re-expression rescues NMDA/AMPA ratios in IRSp53-cKO cortical neurons**. Abnormally heightened NMDAR function was previously observed in hippocampal CA1 pyramidal neurons of mice globally lacking IRSp53[29], although layer 5 pyramidal neurons in the mPFC of IRSp53-cKO mice showed normal NMDAR function[31]. Given that the mPFC has been strongly implicated in social regulation[47–51], we tested if NMDAR functions were altered by adult IRSp53 expression/re-expression in mPFC layer 5 pyramidal neurons of WT and IRSp53-cKO mice.

To this end, we injected PHP.eB-IRSp53 into the retro-orbital sinus of adult WT and IRSp53-cKO mice at postnatal week 8, and then measured the evoked NMDAR- and AMPA receptor

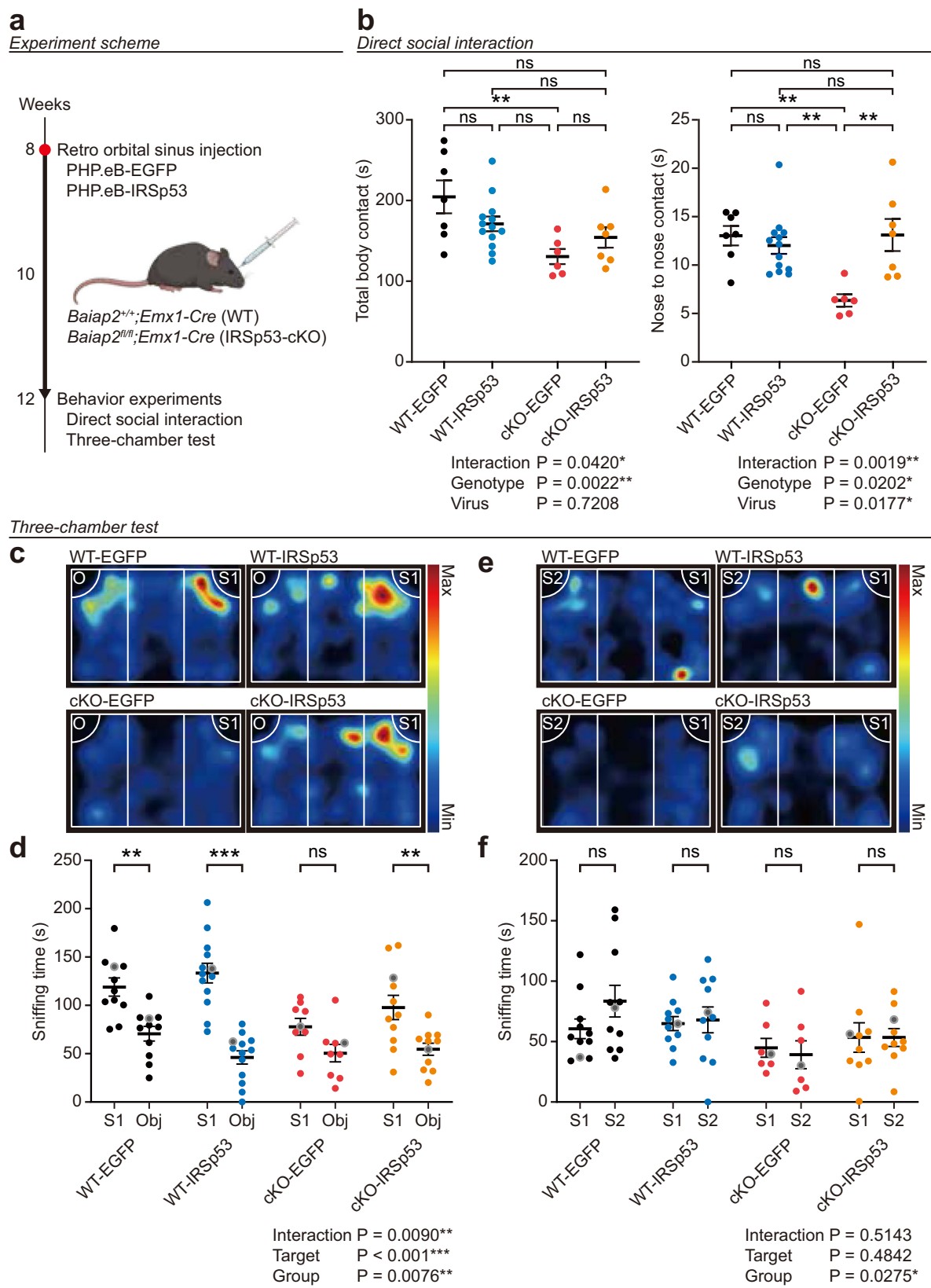

(AMPAR)-mediated excitatory postsynaptic currents (EPSCs) and their ratios in layer 5 pyramidal neurons from the prelimbic region of the mPFC at postnatal week 12 (Fig. 4a, b).

Intriguingly, EGFP-expressing WT and EGFP-expressing IRSp53-cKO neurons showed a significant genotype difference, with IRSp53-cKO neurons displaying increased ratios of

NMDAR- and AMPAR-EPSCs (Fig. 4c, d). This difference contrasted with the lack of change in the NMDA/AMPA ratio in the same neurons reported in a previous study[31], potentially because the mice used in the present study were older (~12 weeks) than those used in the previous study (7–8 weeks). In support of this hypothesis, we found that naive IRSp53-cKO mice at the ages

**Fig. 2 Adult IRSp53 re-expression rescues direct social interaction and social approach in IRSp53-cKO mice. a** Experimental scheme showing injection of PHP.eB-IRSp53 or control (PHP.eB-EGFP) into the retro-orbital sinus of WT or IRSp53-cKO mice at postnatal week 8 and performance of behavioral tests (direct social interaction and three-chamber test) at postnatal week ~12. **b** PHP.eB-IRSp53 injection rescues social interaction in IRSp53-cKO mice in the direct social-interaction test, as shown by nose-to-nose interaction (but not by body contact time), without affecting direct social interaction in WT mice. Mouse pairs of the same genotype (WT or cKO) were used ($n = 7$ mice [WT-EGFP], 13 [WT-IRSp53], 6 [IRSp53-cKO-EGFP], and 7 [IRSp53-cKO-IRSp53], two-way ANOVA with Sidak's test). **c–f** PHP.eB-IRSp53 injection rescues social approach in IRSp53-cKO mice, but not in WT mice, in the three-chamber test, as shown by time spent sniffing stranger 1 (S1) and object (O). Note that the effect of PHP.eB-IRSp53 injection on social novelty recognition (S1 vs. new stranger (S2)) in IRSp53-cKO mice could not be determined because WT mice expressing EGFP (positive control) did not show a baseline preference for S2 over S1 ($n = 11$ mice [WT-EGFP], 13 [WT-IRSp53], 9 [IRSp53-cKO-EGFP], and 11 [IRSp53-cKO-IRSp53] for social approach, 11 mice [WT-EGFP], 11 [WT-IRSp53], 7 [IRSp53-cKO-EGFP], and 10 [IRSp53-cKO-IRSp53] for social novelty, two-way ANOVA [for group × target interactions where group means the four groups [WT-EGFP/IRSp53 and IRSp53-cKO-EGFP/IRSp53] and target means the two targets [S1/O]], Wilcoxon/Student's *t*-test [for the comparison of two different targets [S1 vs. O; S1 vs. S2] within each group [i.e., WT-EGFP], specific *p* values for S1 vs. S2 for each group: 0.3652 [WT-EGFP], 0.7220 [WT-IRSp53], 0.6963 [cKO-EGFP], and 0.8457 [cKO-IRSp53] (see Supplementary Data 1 for further details). Significance values are indicated as \**p* < 0.05, \*\**p* < 0.01, \*\*\**p* < 0.001, or ns (not significant, *p* > 0.05). Error bars represent the standard errors of means.

of 5–6 and 8 weeks showed no genotype difference in the NMDA/AMPA ratio whereas older mutant mice (12 weeks) showed increased NMDA/AMPA ratio (Supplementary Fig. 3). Adult IRSp53 overexpression in WT mice induced an increase in the NMDAR/AMPAR-EPSC ratio (Fig. 4c, d).

As some of these results could be explained by changes in the number of silent synapses, known to mainly contain NMDARs[52–54], we analyzed silent synapses in IRSp53-expressing WT neurons and IRSp53-re-expressing IRSp53-cKO neurons. The results indicated an increase in silent synapses in IRSp53-overexpressing WT but not in IRSp53-re-expressing IRSp53-cKO layer 5 pyramidal neurons (Fig. 4e, f).

We also tested if the changes in NMDA/AMPA ratios and silent synapses may involve changes in the presynaptic release, although there was no genotype difference in the paired-pulse ratios in naive WT and IRSp53-cKO neurons at ~8 weeks[31]. The results indicated an increase in the paired-pulse ratio in IRSp53-overexpressing WT neurons but a decrease in the paired-pulse ratio in IRSp53-re-expressing IRSp53-cKO neurons at 12 weeks (Fig. 4g–j).

These results collectively suggest that IRSp53-cKO neurons at ~12 weeks with IRSp53 re-expression started at ~8 weeks show decreased NMDA/AMPA and paired-pulse ratios and unaltered silent synapses, as compared with control IRSp53-cKO neurons. In addition, WT neurons at ~12 weeks with IRSp53 over-expression started at ~8 weeks show increased NMDA/AMPA and paired-pulse ratios and increased silent synapses, as compared with control WT neurons.

**Adult IRSp53 re-expression decreases the frequency of excitatory synaptic transmission in IRSp53-cKO mice.** The above-described rescue of the NMDA/AMPA ratio in IRSp53-cKO mice by adult IRSp53 re-expression involved a concomitant increase in presynaptic release, which may also alter the frequency of excitatory synaptic transmission. In addition, the decreased pre-synaptic release observed in IRSp53-overexpressing WT neurons may also alter the frequency of excitatory synaptic transmission. To this end, we measured miniature excitatory postsynaptic currents (mEPSCs) in IRSp53-overexpressing WT and IRSp53-re-expressing IRSp53-cKO layer 5 pyramidal neurons.

An analysis of mEPSCs in layer 5 pyramidal neurons of the prelimbic region of the mPFC in WT and IRSp53-cKO mice at 12 weeks injected with PHP.eB-IRSp53 or PHP.eB-EGFP (control) at 8 weeks indicated that there was no difference in the baseline frequency and amplitude between EGFP-expressing WT and IRSp53-cKO mice, as supported by the lack of a genotype difference in two-way ANOVA (Fig. 5a–c). This result differs from that of a previous study showing that mEPSC frequency was decreased in naive IRSp53-cKO mice at ~3 months

(12 weeks)[31]. This difference could be attributable to the use of virus injection (a non-naive condition) in the present study, or a difference in the nature of the utilized controls (*Baiap2*[fl/fl] in the previous study and *Emx1-Cre* mice in the present study). An analysis of miniature inhibitory postsynaptic currents (mIPSCs) indicated no baseline genotype difference (Fig. 5d–f), similar to the previous results[31].

Intriguingly, we herein found that PHP.eB-IRSp53 induced a decrease in the frequency, but not the amplitude, of mEPSCs in both WT and IRSp53-cKO mice, as supported by the finding of a positive virus effect (Fig. 5a–c). In addition, PHP.eB-IRSp53 induced an increase in the amplitude, but not the frequency, of mIPSCs in both WT and IRSp53-cKO mice, as supported by a positive virus effect (Fig. 5d–f). These results suggest that IRSp53 overexpression in WT mice and IRSp53 re-expression in IRSp53-cKO mice induce similar decreases in excitatory synaptic transmission and similar increases in inhibitory synaptic transmission.

## Discussion

Our study demonstrates that adult IRSp53 re-expression by BBB-penetrant PHP.eB-mediated gene delivery can lead to the restoration of NMDAR function and social behavior in IRSp53-cKO mice. These results suggest that adult IRSp53 re-expression has clinical potential for IRSp53-related brain disorders.

Our findings indicate that adult IRSp53 re-expression in the brains of IRSp53-cKO mice after the completion of brain development can normalize synaptic and behavioral deficits induced by IRSp53 deletion. The synaptic normalization of the NMDA/AMPA ratio could be attributable to a decrease NMDAR-EPSCs rather than an increase in AMPAR-EPSCs because we did not observe a change in the amplitude of mEPSCs, which is correlated with levels of AMPARs in functioning excitatory synapses.

Regarding the underlying mechanisms for the normalization of the NMDA/AMPA ratio, global IRSp53 deletion in mice has been shown to cause NMDAR hyperfunction by abnormally stabilizing actin filaments and suppressing activity-dependent actin depolymerization, thereby impairing the long-term depression of NMDARs[29]. The heightened stability of actin filaments in mutant dendritic spines is thought to involve compensatory activation of Rac1 signaling pathways, known to regulate actin polymerization and maintain dendritic spine structures[20]. The activation of Rac1 signaling pathway seems to be triggered by the decrease in the number of dendritic spines in mPFC pyramidal neurons induced by the deletion of IRSp53, considering the known functions of IRSp53 in the promotion of actin polymerization in dendritic spines[20].

The results from the current and previous studies suggest a hypothetical sequence of primary and second events that can

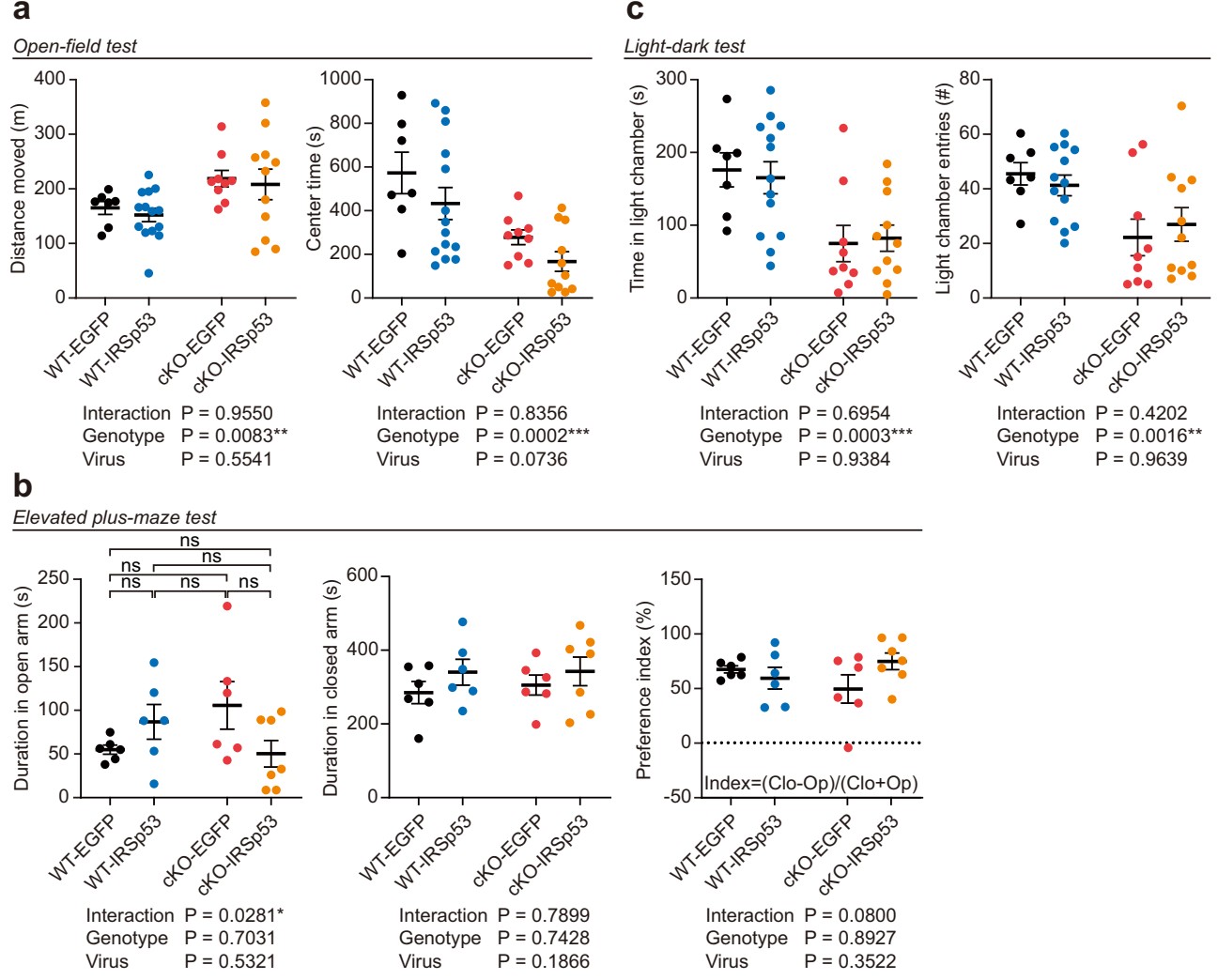

**Fig. 3 Adult IRSp53 re-expression does not affect hyperactivity or anxiety-like behavior in IRSp53-cKO mice. a** PHP.eB-IRSp53 injection does not affect hyperactivity or the time spent in the center region of the open-field arena (a measure of anxiety-like behavior) in adult IRSp53-cKO or WT mice (8–12 weeks), as shown by the lack of a significant virus effect. Note that EGFP-expressing IRSp53-cKO mice show increased baseline hyperactivity and anxiety-like behavior compared to EGFP-expressing WT mice (7 mice [WT-EGFP], 14 [WT-IRSp53], 9 [IRSp53-cKO-EGFP], and 11 [IRSp53-cKO-IRSp53], two-way ANOVA). **b** PHP.eB-IRSp53 injection does not affect anxiety-like behavior in adult IRSp53-cKO or WT mice in the elevated plus-maze test, as shown by the lack of a virus effect on the time spent in open/closed arms and the preference index ([time in closed arms – time in open arms]/total time in closed and open arms). Note that there is no baseline genotype-related difference in these parameters (n = 6 mice [WT-EGFP], 6 [WT-IRSp53], 6 [IRSp53-cKO-EGFP], and 7 [IRSp53-cKO-IRSp53], two-way ANOVA with Sidak's test [duration in open arm]). **c** PHP.eB-IRSp53 injection does not affect anxiety-like behavior in adult IRSp53-cKO or WT mice in the light-dark test, as shown by the lack of a virus effect on the time spent in the light chamber and the light-chamber entry number. Note that EGFP-expressing IRSp53-cKO mice show decreases in these parameters compared to EGFP-expressing WT mice, suggestive of baseline anxiety-like behavior in the former (n = 7 mice [WT-EGFP], 13 [WT-IRSp53], 9 [IRSp53-cKO-EGFP], and 11 [IRSp53-cKO-IRSp53], two-way ANOVA). Significance values are indicated as *p < 0.05, **p < 0.01, ***p < 0.001, or ns (not significant, p > 0.05). Error bars represent the standard errors of means.

occur across postnatal developmental stages in the mutant mPFC. IRSp53 deletion in layer 5 pyramidal neurons may inhibit the normal actin polymerization steps that are required for the maturation and maintenance of dendritic spines and excitatory synapses up until 12 weeks, likely including the stage of ~8 weeks, as supported by the decreased mEPSC frequency at ~12 weeks[31] (see a working hypothesis in Supplementary Fig. 4). The decrease in spine density at ~8 weeks (a primary change) may cause compensatory activation in Rac1 signaling, abnormal stabilization of F-actin in dendritic spines, and NMDAR hyperactivity (secondary changes) during 8–12 weeks, as supported by normal NMDA/AMPA ratio at ~8 weeks but increased NMDA/AMPA ratio at ~12 weeks in naive neurons (Supplementary Fig. 3). In

contrast, re-expression of IRSp53 in mutant neurons starting at ~8 weeks may prevent the changes in Rac1 signaling, F-actin, and NMDAR activity from occurring. However, IRSp53 re-expression (8–12 weeks) seems to fail to normalize the reduced spine density, as supported by the reduced mEPSC frequency in IRSp53-re-expressing mutant neurons, compared with control (EGFP alone) mutant neurons, at ~12 weeks[31]. In addition, this failure accompanied an increase in presynaptic release in mutant neurons (Fig. 4i, j), which may represent a compensatory change to overcome the decreased excitatory synaptic transmission at mutant synapses.

Our previous studies suggested distinct synaptic changes in the hippocampus and mPFC in IRSp53-cKO mice; normal synapse

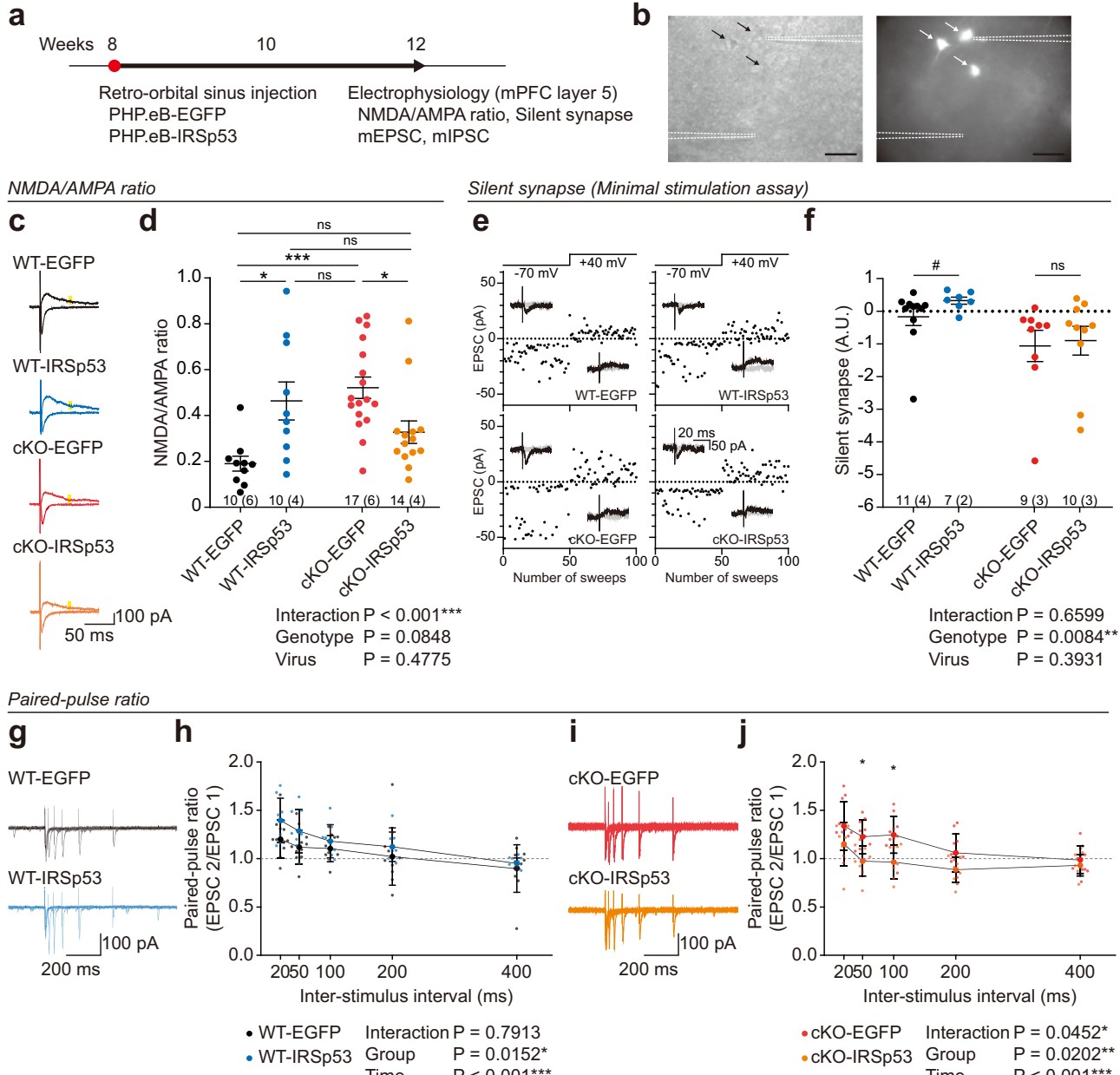

**Fig. 4 Adult IRSp53 re-expression rescues NMDA/AMPA ratios in IRSp53-cKO cortical neurons. a** Experimental scheme showing injection of PHP.eB-IRSp53 or control (PHP.eB-EGFP) into the retro-orbital sinus of WT or IRSp53-cKO mice at postnatal week 8 and electrophysiological measurements (evoked and miniature EPSCs) at postnatal week ~12. **b** Examples of layer 5 pyramidal neurons in the prelimbic region of the mPFC, where NMDAR/AMPAR-EPSCs or mEPSCs were measured, from WT mice injected with PHP.eB-EGFP into the retro-orbital sinus at postnatal week 8 and visualized under fluorescence microscopy at postnatal week 12 without additional EGFP staining. The EGFP signals in the neurons were enhanced to clearly indicate the neurons analyzed by patch-clamp recordings. **c** Sample traces of evoked NMDAR- and AMPAR-EPSCs from EGFP/IRSp53-expressing WT/IRSp53-cKO mice. **d** PHP.eB-IRSp53 injection normalizes the ratio of NMDAR-EPSCs and AMPAR-EPSCs in mPFC layer 5 pyramidal neurons of adult IRSp53-cKO mice. Note that IRSp53 overexpression in WT mice increases the NMDA/AMPA ratio ($n = 10$ neurons from 6 mice [WT-EGFP], 10, 4 [WT-IRSp53],17, 6 [IRSp53-cKO-EGFP], and 14, 4 [IRSp53-cKO-IRSp53], two-way ANOVA with Sidak's test). **e, f** IRSp53 overexpression in WT neurons increases silent synapses, whereas IRSp53-re-expression in IRSp53-cKO neurons does not affect silent synapses ($n = 11$, 4 [WT-EGFP], 7, 2 [WT-IRSp53], 9, 3 [IRSp53-cKO-EGFP], and 10, 3 [IRSp53-cKO-IRSp53], two-way ANOVA with Sidak's test, #$p < 0.05$ [Mann-Whitney test]). **g, h** Increased paired-pulse facilitation (PPF) in IRSp53-overexpressing WT mPFC layer 5 pyramidal neurons. WT mice were infected with PHP.eB-IRSp53, or PHP.eB-EGFP (control), at 8 weeks, and EGFP-positive neurons were used to measure PPF at 12 weeks ($n = 11$, 2 [WT-EGFP], 8, 2 [WT-IRSp53], two-way ANOVA with Sidak's test).
**i, j** Decreased PPF in IRSp53-re-expressing IRSp53-cKO mPFC layer 5 pyramidal neurons. IRSp53-cKO mice (Emx1) were infected with PHP.eB-IRSp53, or PHP.eB-EGFP (control), at 8 weeks, and EGFP-positive neurons were used to measure PPF at 12 weeks ($n = 11$, 2 [cKO-EGFP], 7, 2 [cKO-IRSp53], two-way ANOVA with Sidak's test). Significance values are indicated as *$p < 0.05$, **$p < 0.01$, ***$p < 0.001$, or ns (not significant, $p > 0.05$). Error bars represent the standard errors of means.

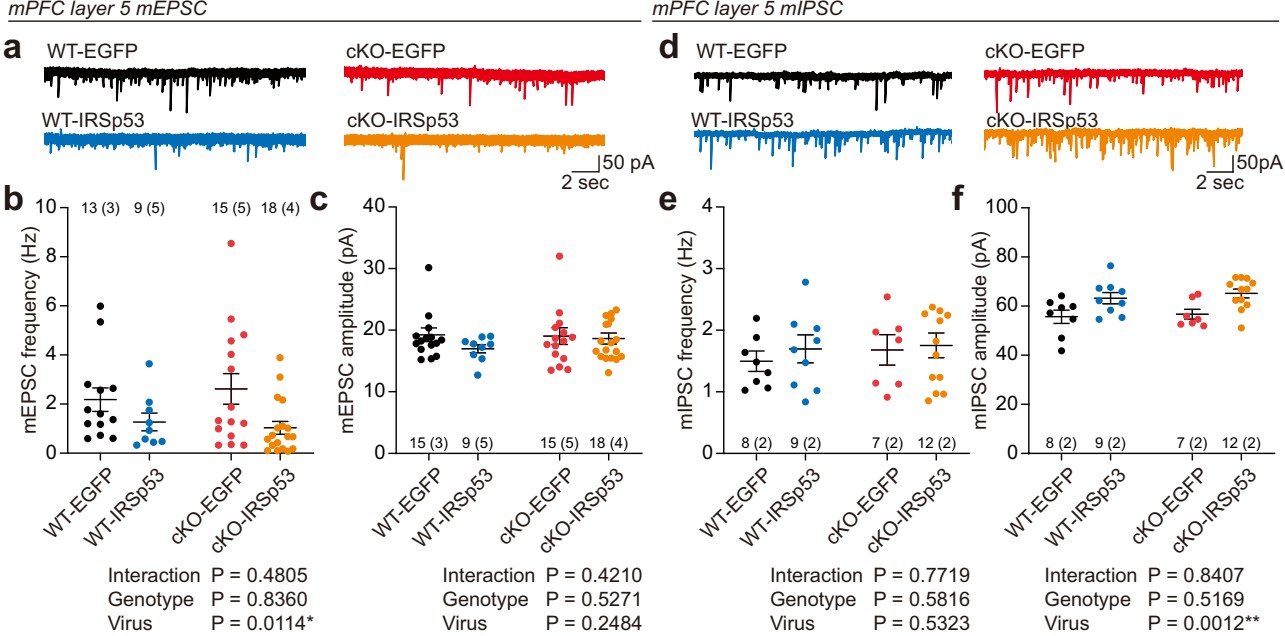

**Fig. 5 Adult IRSp53 re-expression decreases excitatory synaptic transmission and increases inhibitory synaptic transmission in IRSp53-cKO mice.**
**a** Examples of mEPSC traces in layer 5 pyramidal neurons from the prelimbic region of the mPFC in WT and IRSp53-cKO mice that received retro-orbital sinus injections of PHP.eB-IRSp53 or PHP.eB-EGFP (control) at postnatal week 8 and underwent mEPSC measurements at postnatal week 12.
**b, c** Decreased frequency, but not amplitude, of mEPSCs in layer 5 pyramidal neurons of WT and IRSp53-cKO mice injected with PHP.eB-EGFP or PHP.eB-IRSp53 (n = 15 neurons from 3 mice [WT-EGFP], 15, 4 [WT-IRSp53], 9, 5 [IRSp53-cKO-EGFP], and 18, 4 [IRSp53-cKO-IRSp53], two-way ANOVA).
**d** Examples of mIPSC traces in layer 5 pyramidal neurons from the prelimbic region of the mPFC in WT and IRSp53-cKO mice that received retro-orbital sinus injections of PHP.eB-IRSp53 or PHP.eB-EGFP (control) at postnatal week 8 and underwent mIPSC measurements at postnatal week 12. **e, f** Increased amplitude, but not frequency, of mIPSCs in layer 5 pyramidal neurons of WT and IRSp53-cKO mice injected with PHP.eB-EGFP or PHP.eB-IRSp53 (n = 8, 2 [WT-EGFP], 9, 2 [WT-IRSp53], 7, 2 [IRSp53-cKO-EGFP], and 12, 2 [IRSp53-cKO-IRSp53], two-way ANOVA). Significance values are indicated as *p < 0.05, **p < 0.01, ***p < 0.001, or ns (not significant, p > 0.05). Error bars represent the standard errors of means.

density and increased NMDAR function in the hippocampus and decreased synapse density and normal NMDAR function in the mPFC[28,29,31]. The results from the current study suggest an updated view that both hippocampus and mPFC may undergo similar primary and secondary changes but over different temporal trajectories; secondary changes occur during 8–12 weeks in the mPFC and during 1–3 weeks in the hippocampus. Testing this possibility would have to involve, i.e., testing the changes in spine density and synaptic functions at an early postnatal stage in the hippocampus.

Intriguingly, IRSp53-overexpressing WT neurons show an increase in the NMDA/AMPA ratio (Fig. 4c, d). This could be attributable to an increase in NMDAR-EPSCs rather than a decrease in AMPAR-EPSCs, as suggested by the lack of change in the mEPSC amplitude. Regarding potential mechanisms underlying the increase in NMDAR-EPSCs, the acute up-regulation of IRSp53 proteins in WT neurons seems to promote the generation of silent excitatory synapses (Fig. 4e, f), known to represent nascent synapses mainly containing NMDARs[52–54], likely through membrane-deforming/protrusive effects of IRSp53 and excessive F-actin polymerization at nascent synapses[20] as well as enhanced F-actin-dependent NMDAR clustering and activation[55–57] (see a working hypothesis in Supplementary Fig. 4). This may induce the redistribution of presynaptic proteins from existing presynaptic sites to new silent synapses, as supported by the increased paired-pulse facilitation at presynaptic sites of IRSp53-overexpressing WT neurons (Fig. 4g, h). This might also accompany the redistribution of postsynaptic proteins from existing excitatory synapses to silent synapses and a decrease in the number of functional excitatory synapses, which may be reflected in the decreased mEPSC frequency in IRSp53-overexpressing WT neurons. However, we

should point out that these predicted events are highly speculative. One way to test this hypothesis is to perform an electron microscopic analysis of WT neurons overexpressing IRSp53 to see if the number of NMDAR-only silent synapses was indeed increased and the number of pre-existing synapses containing both NMDARs and AMPARs was decreased.

Our results showed that adult IRSp53 re-expression rescues social interaction but not hyperactivity or anxiety-like behavior in IRSp53-cKO mice, reminiscent of social but not hyperactivity rescue in IRSp53-global KO mice treated with memantine (NMDAR antagonist) and MPEP (mGluR5 antagonist)[29]. It is possible that the neural circuits associated with hyperactivity or anxiety-like behavior in IRSp53-cKO mice may differ from those associated with social interaction, such that IRSp53 is more important for social interaction-related synapses and neural circuits at adult stages. Related brain regions of IRSp53 re-expression could be the mPFC as well as other social-related brain regions, as evident from the widespread EGFP signals in the IRSp53-re-expressing IRSp53-cKO brain. Alternatively, it is possible that social interaction-related neural circuits develop at time points later than those for locomotor- and anxiety-related behaviors, assuming that the neural circuits that are more recently developed are more likely to be rescued than those formed at more remote time points. Lastly, it is possible that neural circuits involved in social interaction could be generally more rescuable than those involved in locomotion and anxiety-like behaviors, perhaps being more flexible and susceptible to synaptic modulations.

Gene re-expression by the BBB-penetrant PHP.eB approach has clinical potential to be used in treating cases with loss-of-function gene mutations. In the context of attempting related

experiments in animal models, one open question is whether virus-mediated adult gene re-expression will drive relevant gene expression in all brain cell types that endogenously express the missing gene. This issue did not need to be considered in previous mouse studies using gene flexing and tamoxifen- and Cre-dependent gene restoration, which can occur easily in all cell types. Our study used Emx1-dependent gene deletion and re-expression restricted to excitatory neurons in the cortex and hippocampus, solving the issue of re-expressing the IRSp53 gene in correct cell types. Notably, researchers recently developed the mCREATE system, through which AAV variants with distinct target cell tropisms can be used to drive gene expression in distinct cell types, such as neuronal or glial cell types[58].

Another clinically important question is whether the gene re-expression at adult stages will be sufficient for phenotypic restoration, or if earlier gene restoration at early postnatal or juvenile stages will prove necessary. Our study reveals that adult IRSp53 re-expression rescues social interaction but not hyperactivity or anxiety-like behavior in IRSp53-cKO mice. These results are largely similar to those from Shank3- and SynGAP1-mutant mice but not those from MeCP2- and GluN1-mutant mice[3–14]. Therefore, converging features with regard to the optimal timing of gene re-expression seem to require additional studies on other genes, which is an important issue for potential gene-based therapy[2].

IRSp53 has been implicated in neuropsychiatric disorders, including ASD[34,35], schizophrenia[36,37], and ADHD[38,39]. The results of the present study suggest that it may be possible to use the PHP.eB system and adult IRSp53 re-expression in the treatment of IRSp53-related brain disorders and, more broadly, other brain disorders that can benefit from the re-expression of target genes via BBB penetration. Given that somatic gene editing is becoming increasingly feasible[59], related machinery could be loaded into the PHP.eB system.

The current study uses a mouse system to drive adult IRSp53 re-expression. It remains to be determined if this approach will also work for the human brain or human brain organoids. In addition, as the current study limited IRSp53 re-expression to Emx1-positive excitatory neurons using genetic methods (Emx1-Cre mice and PHP.eB-DIO-IRSp53), future approaches involving human and genetically non-modified brain cells would need to consider ways to ensure that the expression of IRSp53 takes place specifically in brain cells that endogenously express IRSp53. Lastly, as we did not optimize our IRSp53 expression to mimic the endogenous expression levels in the present work, follow-up studies should also consider ways to address this aspect.

In summary, our results indicate that BBB-penetrant PHP.eB can be used to drive adult IRSp53 re-expression in defined cell types in the mouse brain and that this can normalize synaptic and behavioral deficits in IRSp53-cKO mice, suggesting that PHP.eB-mediated adult IRSp53 re-expression has therapeutic potential.

## Methods

**Animals**. Mice carrying exons 4–6 of the Baiap2 gene, encoding IRSp53, floxed by LoxP (Baiap2fl/fl) in a C57BL/6J background were designed and generated by Biocytogen. We crossed these IRSp53-floxed mice with Emx1-Cre mice (JAX #005628) to generate Emx1-Cre;Baiap2fl/fl mice (termed IRSp53 conditional knockout or cKO).

For CaMKIIα-Cre-dependent IRSp53 deletion, we crossed Baiap2fl/fl mice with CaMKIIα-Cre mice (JAX #005359)[46]. Mouse genotyping was performed by polymerase chain reaction (PCR) using the following PCR primers: IRSp53-floxed allele, 5'-AGGAGGTGTTTCTGCTCTGG-3'/5'-AATAGCAGTCTGGGGTCTGG-3'; Cre-positive allele, 5'-CGTACTGACGGGTGGGAGAAT-3'/5'-TGCATGATCTCCGGTATTGA-3'. The animal study was reviewed and approved by the Committee of Animal Research at KAIST (KA2020-94). Mice were housed and bred at the mouse facility of KAIST and maintained according to the Animal Research Requirements of KAIST. All mice were fed ad libitum and caged under a 12-h light/dark cycle. Mice were weaned at postnatal day 21, and two to six mixed-genotype littermates were housed together until experiments. There was no difference in the body weights of age-matched mouse groups.

**Virus packaging and titration**. The pAAV-hSyn1-DIO-mEGFP construct was courtesy of Dr Bryan Roth (Addgene #50457). Human IRSp53 cDNA with an N-terminal HA tag was subcloned into pAAV-hSyn1-DIO-mEGFP to produce pAAV-hSyn1-DIO-mEGFP-P2A-HA-IRSp53 construct. AAVs (serotype PHP.eB) packaging was performed as described previously[60]. HEK293T cells at 90% confluency were used for the transfection. AAV-DIO-IRSp53 (10 μg), PHP.eB plasmid (20 μg), and pHelper plasmid (20 μg) were co-transfected (150 mm plates) using the polyethyleneimine solution (Polysciences) method. The culture media at 72 and 120 h after transfection were collected, added with 40% PEG 8000 (5× solution; Sigma, 89510), and centrifuged at 4000 × g for 30 min for protein precipitation. The pellets and the cell lysates (collected at 120 h) were suspended in SAN digestion solution (100 U/mL HL-SAN (Heat-Labile Salt-Activated Nuclease), Arcticzymes; SAN digestion buffer: 25 mM Tris-HCl, pH 8.5, 5 mM MgCl2, 0.5 M NaCl). Resuspension solution were collected and centrifuged on iodixanol (Opti-prep, Sigma) gradient consist of 60%, 40%, 25%, and 15% (Beckman coulter, 361625). After the centrifugation at 350,000 × g for 2 h at 4 °C, the virus solution in the Optiprep gradient of 40–60% was collected, diluted with phosphate-buffered saline (GIBCO), and concentrated using Amicon ultrafilter (100 K; Merck). The final virus solution was stored at −80 °C until use. Virus titration was conducted using a qPCR method and SYBR green (Elpis, EBT-1801) master mix and CFX96 real-time system (Bio-Rad). The following primer sets for WPRE sequence were used for qPCR quantification; forward, 5'-TCTTTATGAGGAGTTGTGGCCCGT-3', reverse, 5'-ACAACACCACGGAATTGTCAGTGC-3'.

**Retro-orbital sinus injection**. For intravascular delivery of AAVs, retro-orbital sinus injection was used[61]. Mice (~8 weeks, male) were placed in a Plexiglas chamber filled with 4% v/v (volume-to-volume) isoflurane and 4% isoflurane for anesthesia. Virus solution (200 μL, ~10^11 vg/mL) was injected into the right retro-orbital sinus of the mouse using an insulin syringe. During the injection, 2% v/v isoflurane was used to maintain anesthesia.

**Electrophysiology**. For electrophysiological experiments, N-Methyl-D-glucamine (NMDG)-artificial cerebrospinal fluid (NMDG-ACSF) buffer-based protective recovery method was used. Mice at 3 months were anesthetized with isoflurane (Terrell solution, Piramal Healthcare), followed by cardiac perfusion of NMDG-ACSF buffer containing the following (in mM): 100 NMDG, 12 N-acetyl-L-cysteine (NAC), 30 NaHCO3, 20 HEPES, 25 glucose, 2 thiourea, 5 Na-ascorbate, 3 Na-pyruvate, 2.5 KCl, 1.25 NaH2PO4, 0.5 CaCl2, and 10 MgSO4, at room temperature. Mouse brains were extracted, and sectioned (300 μm, Leica VT1200) in the abovementioned NMDG-ACSF buffer, at room temperature. Brain slices were transferred to an NMDG-ACSF containing chamber at 32 °C, for 11 min, and a HEPES buffer containing recovery chamber at room temperature, for 1 h. HEPES buffer containing follow (in mM): 92 NaCl2, 12 NAC, 30 NaHCO3, 20 HEPES, 25 glucose, 2 thiourea, 5 Na-ascorbate, 3 Na-pyruvate, 2.5 KCl, 1.25 NaH2PO4, 2.5 CaCl2, and 1.3 MgCl2 oxygenated with 95% O2 and 5% CO2 gases.

Whole-cell patch-clamp recordings of mPFC layer 5 pyramidal neurons were made using a Multiclamp 700B amplifier (Molecular Devices) and Digidata 1550 (Molecular Devices). Series resistance was monitored during each whole-cell patch-clamp recording, by measuring the peak amplitude of the capacitance currents in response to a short hyperpolarizing step pulse (5 mV, 40 ms); only data with a change less than 20% were used. Glass pipettes for stimulation and recording were made using borosilicate glass capillaries (Harvard Apparatus) and a micropipette puller (Narishige).

For mEPSC and mIPSC experiments, mice at ~3 months (4 weeks after retro-orbital sinus injection) mPFC slices were used. For mEPSC experiments, TTX (1 μM, Abcam) and picrotoxin (100 μM, Sigma) were added to ACSF. Holding potential for mEPSC was −70 mV, and recording pipettes (3.0–4.0 MΩ) were filled with an internal solution containing (in mM) 100 CsMeSO4, 10 TEA-Cl, 8 NaCl, 10 HEPES, 5 QX-314-Cl, 2 Mg-ATP, 0.3 Na-GTP, and 10 EGTA, with pH 7.25, 295 mOsm. For mIPSC experiments, TTX (1 μM), NBQX (10 μM, Tocris), and D-AP5 (50 μM, Hello bio) were added to ACSF. Holding potential for mIPSC was −70 mV, and recording pipettes (3.0–4.0 MΩ) were filled with an internal solution containing (in mM) 120 CsCl, 10 TEA-Cl, 8 NaCl, 10 HEPES, 5 QX-314-Cl, 4 Mg-ATP, 0.3 Na-GTP, and 10 EGTA, with pH 7.35, 280 mOsm.

For NMDA/AMPA ratio experiments, picrotoxin (100 μM) was added to circulating ACSF to inhibit IPSCs. Stimulation pipettes were filled with normal ACSF. Recording pipettes (3.0–4.0 MΩ) were filled with the abovementioned internal solution for mEPSC experiments. A stimulation pipette was located ~150 μm toward the pia from the recording pipette. Pyramidal neurons were voltage-clamped at −70 mV to measure AMPAR-EPSCs. Electric stimulations for EPSC evoke were given at every 15 s, and 20 consecutive responses were recorded. After recording AMPAR-EPSCs, the holding potential was changed to +40 mV to record NMDAR-EPSCs, and also 20 responses were recorded. NMDAR-EPSCs were measured at 50 ms after the stimulation. The NMDA/AMPA ratio was determined by dividing the mean value of 20 NMDAR-EPSC amplitudes (50 ms) by the mean value of 20 AMPAR-EPSC peak amplitudes.

For silent synapse and paired-pulse ratio measurements, the conditions identical to NMDA/AMPA ratio experiments (circulating ACSF, internal solution, and stimulation condition) were used. The minimal stimulation assay was performed to measure silent synapses, as described previously[62]. After obtaining

AMPAR-EPSCs (EPSC >200 pA, and <10 pA at 50 ms after stimulation) at −70 mV, the stimulation intensity was reduced to obtain small (EPSCs <50 pA) and stochastically failing (~50%) AMPAR-EPSCs. The stimulation intensity was kept constant for the rest of the recordings. For each cell, 50 sweeps were recorded at −70 mV, and the following 50 sweeps were recorded at +40 mV. We visually detected failed traces in a blind, unbiased manner. The level of silent synapses was estimated using the formula: $1 − \ln(F_{−70})/\ln(F_{+40})$, $F_{−70}$ was the failure rate at −70 mV and $F_{+40}$ was the failure rate at +40 mV. For paired-pulse ratio experiments, stimulation intensity was modified as; 60 pA < EPSC1 < 200 pA. For each cell, 10 sweeps were recorded for each inter-stimulus interval (ISI, 20, 50, 100, 200, and 400 ms) at –70 mV.

**Western blot**. Brains from mice at ~3 months (4 weeks after virus injection for cKO-IRSp53 mice) were extracted. Cortical and hippocampal regions were polled and collected in ice-cold homogenization buffer (0.32 M sucrose, 10 mM HEPES, 2 mM EDTA, 2 mM EGTA, 50 mM NaF, 1 mM sodium orthovanadate, and protease inhibitors) and homogenized by tissue grinder. The following antibodies were purchased; BAIAP2/IRSp53 antibody (Atlas, rabbit, HPA023310, 1:4000), GAPDH antibody (Cell Signaling, mouse, #97166, 1:4000). Primary antibodies were incubated with membranes overnight at 4 °C. Odyssey Fc Imaging System was used to detect secondary antibody signals.

**Immunohistochemistry**. Mice at ~3 months (4 weeks after virus injection) were anesthetized with isoflurane and subjected to cardiac perfusion with ~50 mL of phosphate-buffered saline and 4% paraformaldehyde (PFA) consecutively before brain extraction. Extracted brains were post-fixed in 4% PFA solution at 4 °C, overnight. Fixed brains were sectioned into slices with 50-μm thickness using a vibratome (Leica, VT1200). The following antibodies were used for immunohistochemistry: IRSp53 (Atlas, rabbit, 1:500), HA (Sigma, ms, 1:500), anti-rabbit 647 (Jackson, 1:500), anti-mouse 647 (Jackson, 1:500). Mounted sections were imaged using an LSM 780 confocal microscope (Zeiss).

**Fluorescent in situ hybridization**. In brief, frozen mouse brain sections (14-μm thick) were cut coronally through the mPFC and primary motor area, and thaw-mounted onto Superfrost Plus Microscope Slides (Fisher Scientific). The sections were fixed in 4% PFA, followed by dehydration in increasing concentrations of ethanol and proteinase digestion. For hybridization, the sections were incubated in different amplifier solutions in a HybEZ hybridization oven (Advanced Cell Diagnostics) at 40 °C. The probes used in these studies were three synthetic oligonucleotides complementary to the nucleotide sequence 2-1268 of Mm-IRSp53-C1, 464–1415 of Mm-Slc17a7 (Vglut1)-C2, and 1986–2998 of Mm-Slc17a6 (Vglut2)-C3 (Advanced Cell Diagnostics). The labeled probes were conjugated to Alexa Fluor 488, Atto 550, or Atto 647. The sections were hybridized with probe mixtures at 40 °C for 2 h. Nonspecifically hybridized probes were removed by washing the sections in 1× wash buffer, and the slides were treated with Amplifier 1-FL for 30 min, Amplifier 2-FL for 15 min, Amplifier 3-FL for 30 min, and Amplifier 4 Alt B-FL for 15 min. Each amplifier was removed by washing with 1× wash buffer. The slides were viewed and photographed using TCS SP8 Dichroic/CS (Leica). After FISH was performed, the average number of dots per cell was quantified using the HALO image analysis algorithm in HALO v3.0.4 software (Indica Labs).

**Behavioral assays**. All behavioral assays were performed with age-matched male mice (8–16 weeks). Mice were handled for three days (10 min a day) prior to the application of a battery of behavioral experiments. Mice were allowed to rest for at least 1 day between tests. Behavior tests were conducted in an effort to minimize the stress levels of mice, and were performed during the light-off period.

**Open-field test**. Each mouse was placed in a white matt acryl box (40 × 40 × 40 cm) with a light illumination of ~100 lux. The activity of a freely moving mouse in the chamber was video-recorded for 1 h. The center zone in the open-field apparatus was defined as a square of 20 × 20 cm in the center. The activity of mice including distance moved and duration in the center zone was measured using EthoVision XT13 (Noldus).

**Elevated plus-maze test**. The EPM consists of two open and two closed arms (5 × 30 × 0.5 cm). Closed arms were enclosed by a 30-cm wall without a ceiling. Each arm was attached to the center region (5 × 5 cm) orthogonally and elevated from the floor (75 cm). At the beginning of the test, a mouse was placed in the center region of the maze facing the open arm. The open arm was illuminated at a light intensity of ~300 lux. Mouse movement was recorded for 10 min. Time spent in open/closed arms and frequency of entry to open/closed arms were measured using EthoVision XT13 software (Noldus). The preference index for closed arms (%) was calculated as follows:

$$(Closed − Open)/(Closed + Open) ∗ 100 \tag{1}$$

**Light-dark test**. The light-dark apparatus consists of a white acryl chamber (20 × 30 × 20 cm; light) with an open roof connected to an enclosed, black acryl

chamber (20 × 13 × 20 cm; dark) with an entrance that allows mice to freely move between the two chambers. The light chamber was illuminated at ~300 lux. Mice were placed in the light chamber facing away from the entrance to the dark chamber, and their movements were recorded for 10 min. Entry into the light chamber was counted only when the mouse's entire body crossed the entrance. The time spent in each chamber and the number of entries into the light box were measured manually in a blinded manner.

**Three-chamber test**. The three-chamber test of social interaction and social novelty recognition was performed as described previously[63–65] with slight modification (see below for details). The utilized apparatus (40 × 60 × 22 cm) comprised one center chamber (40 × 20 × 22 cm) and two juxtaposed side chambers (40 × 20 × 22 cm), with a light (intensity, ~100 lux) at the center. Each side chamber contained a metal cage in the corner. The entrances between the center and side chambers allowed mice to freely explore the entire apparatus. The task was composed of three 10-min sessions: empty (E)-empty, stranger 1 (S1)-object (O), and S1-stranger 2 (S2). In the first session, a test mouse was allowed to explore all three chambers, with the metal cages empty. Before the second session, mouse S1 (age- and sex-matched 129/SvJae) and an object (O, here a blue plastic cylinder) were placed in the metal cages, which were chosen randomly to avoid side preference. In the second session, the subject mouse was allowed to explore all three chambers and cages. After the second session, the object was replaced with a novel stranger mouse (S2). In the third session, the subject mouse was allowed to explore S1 and S2 for 10 min. The time spent sniffing each social/object target was measured using the EthoVision XT13 software (Noldus). The preference index (%) was calculated as follows:

$$(S1 − O)/(S1 + O) ∗ 100 \tag{2}$$

for social interaction, and

$$(S2 − S1)/(S2 + S1) ∗ 100 \tag{3}$$

for social novelty recognition.

**Dyadic social interaction**. The direct social interaction test was performed as described previously[29]. Each individual mouse was habituated to a gray box (30 × 30 × 30 cm; ~100 lux illumination at the center) for 2 consecutive days (10 min/day). On day 3, a subject mouse and a stranger WT mouse (age, sex, and genetic background [C57BL/6J] matched) were placed in the test box simultaneously. The activity of the subject mouse was recorded for 10 min. The time spent in nose-to-nose interaction was measured manually in a blinded manner. Nose-to-nose interaction was defined as sniffing the head part of the other mouse. Total body contact time was automatically analyzed by EthoVision XT13.

**Novel object recognition test**. Mice were habituated in the open-field box without objects for 30 min, 24 h before the training session. In the training session, mice were placed in the box with two identical objects (blue plastic cylinders) for 10 min. Twenty-four hours after the training, one of the objects was replaced with a different object (metal rectangular prism). Object recognition was scored by the amount of time spent with the nose of the mouse pointing and located within 2 cm from the object.

**Statistics and reproducibility**. Statistical analyses were performed using GraphPad Prism 9 (GraphPad Software). Differences were considered significant at $p$ values <0.05*, <0.01**, and <0.001***. The exact $n$ numbers and statistical tests used are indicated in the figure legends. For two-way ANOVA, multiple comparisons were performed when interactions were significant. All results are represented as dot plots, with error bars representing the standard error of the mean. Statistical details and exact n numbers for each experiment are described in Supplementary Data 1.

**Reporting summary**. Further information on research design is available in the Nature Research Reporting Summary linked to this article.

## Data availability
Source data underlying figures are presented in Supplementary Data 1. Uncropped blots are presented in Supplementary Fig. 1.

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

## Acknowledgements

We would like to thank Dr. Viviana Gradinaru at Caltech for providing the PHP.eB plasmid and advising on virus packaging. This work was supported by the NRF (National Research Foundation of Korea) Global PhD Fellowship (NRF-2019H1A2A1076692 to Y.W.N.), NRF grant 2020M3E5D9080794 (to H.K.), and the Institute for Basic Science (IBS-R002-D1 to E.K.).

## Author contributions

Y.W.N., C.Y., J.K., and S.L. performed behavioral experiments and electrophysiological experiments; C.Y. performed immunohistochemistry; Y.W.N. and Y.K. performed immunoblot experiments; E.Y. performed in situ hybridization experiments; H.K. and E.K. designed research and wrote the manuscript.

## Competing interests

The authors declare no competing interests.

## Ethical approval

The animal study was reviewed and approved by the Committee of Animal Research at KAIST (KA2020-94). Mice were housed and bred at the mouse facility of KAIST and maintained according to the Animal Research Requirements of KAIST.
