## [Peer Review File · Communications Biology]

Reviewers' comments:

Reviewer #1 (Remarks to the Author):

This manuscript describes adult IRSp53 re-expression in IRSp53-mutant mice via a brain-blood barrier-penetrant AAV retro-orbital injection could rescue the social deficits and NMDAR-mediated excitatory synaptic transmission in the medial prefrontal cortex. Although the data are of high quality, I don't see any new conceptual /technical or mechanistical innovations. The authors found that re-expression IRSp53 in adult period restores IRSp53-dysfunction related phenotypes in mice, while no further mechanism was uncovered in this field. Since restored important genes at adult stages after the completion of brain development to normalize defective brain functions have been reported, such as MeCP2, GluN1, Shank3, SynGAP1, as mentioned in the introduction of this manuscript, there are not sufficient conceptual advances in this manuscript.

Other issues:

1. What's the phenotype of adult deletion of IRSp53. Is it different from the IRSp53 knock-out from embryonic or early postnatal stages?
2. The authors show the restored social deficits in IRSp53-mutant mice by IRSp53 re-expression. The mPFC has been strongly implicated in social regulation, so the authors chose the mPFC to detect NMDAR function, while I don't see obvious EGFP signals representing IRSp53 expression by BBB-penetrant PHP.eB in mPFC from Fig 1C. The restored social deficits found in this manuscript may be attribute to other area.
3. The data show in Fig2e are inconsistent with the statistics in Fig2f. The author didn't observe obvious difference between a novel stranger and a familiar stranger, while the data in Fig2e show an obvious tendentiousness to S2.
4. Both deletion of IRSp53 and adult IRSp53 overexpression in WT mice induced an increase in the NMDAR/AMPA-EPSC ratio, this is a surprising result. It will be interesting to uncover the bidirectional regulation mechanism of IRSp53 in NMDAR functions.

Reviewer #2 (Remarks to the Author):

The manuscript describes a „rescue experiment“ for a neurodevelopmental disorder. Mice lacking the gene coding for the insulin receptor substrate of 53 kDa (IRSp53) are treated with an AAV expression vector to correct expression of this gene, and are analysed on a behavioural as well as electrophysiological level.

IRSp53 deficient mice were originally described with a strong phenotype in learning and memory (as evident in the Morris water maze, and contextual fear conditioning assays). Later on it was also shown that these mice exhibit an autism related phenotype, exemplified by reduced social interactions. On a molecular/cellular level, these findings correlated with an enhanced NMDA receptor function, evident as increased NMDA/AMPA ratio. In humans, variants in the coding gene for this protein correlate in a few cases with neurodevelopmental disorders including autism and schizophrenia.

As these findings pointed to a neurodevelopmental disorder, it was rather interesting to see here whether reexpression of IRSp53 by means of a virus at the adult stage would lead to a proper rescue of the abovementioned phenotypic features. Importantly, conditional ko mice are used which lack IRSp53 expression only in excitatory/glutamatergic neurons, and the reexpression strategy makes sure that only these neurons are targeted by the transgene.

Initially the authors observe efficient expression of the protein, as evidenced by fluorescence microscopy. In behavioural assays, the reexpression of IRSp53 induces an impressive rescue of the social interaction, as evident in “nose-to-nose” contact of the mice (Fig. 2b). On the other hand, IRSp53 reexpression does not rescue the increased locomotor activity (open field assay), or the anxiety phenotype (elevated plus maze).

In electrophysiological experiments the authors claim that IRSp53 reexpression partially rescues the

increased NMDA/AMPA ratio observed in ko mice. On the other hand, they also observed an increase in NMDA/AMPA ratio upon IRSp53 overexpression in wt mice.

Overall this is a very interesting study, as it addresses the important question whether and when it is possible to rescue the neurodevelopmental phenotype through reexpression of the target gene (which is mutated in the patient, or in this case in the mouse).

Nevertheless, there are several problems associated with this manuscript which should be addressed. Major points.

p. 5, bottom, and p.6 top, as well as Fig. 1a: the genetic approach is not well explained; I assume that the triangles in the vector map are LoxP sites; and that a recombination event is supposed to take place in cells where the genomic Baiap2 is knocked out by Cre recombinase. But this is really not evident from the text.

p. 6, and Fig. 1. There is no information regarding the 2A-peptide cleavage. As there appears to be almost perfect colocalization between HA-and GFP signals in Figure 1d, one might conclude that cleavage did not occur. Otherwise the HA-signal should be more distinct from the GFP signal, as HA-IRSp53 is supposed to be synaptically localized. A western blot analysis of affected brain regions would clarify this, and would also provide information regarding the levels of overexpression achieved compared to wt, and cKO animals.

Overall, there is no information regarding the efficiency of the viral approach. Which percentage of cells is infected, in particular which percentage of the glutamatergic/excitatory neurons?

Fig. 2c,d and Fig2e,f. These figures are not well explained. Assuming that the heat maps show representative behaviour of mice, one would assume that cKO-IRSp53 mice spend 90 % of their time with S1 and very little time with object O. The bar graph in 2d then gives a very different impression (a moderate 50 % increase in time spent with S1, compared to time with object O). Then the authors claim that in 2e, wt mice do not show a preference for stranger S2; but in fact the heat map shows very close interaction with S2 for the wt-EGFP positive control, very much in contrast to cKO-EGFP, which stays away from S2 as far as possible. Is the heat map only for a single mouse? Which is not representative for the whole cohort of mice?

Fig. 4. I do not think that, as claimed by the authors, there is a partial rescue of NMDA/AMPA ratio in Fig. 4d. There is no difference between cKO-EGFP and cKO-IRSp53 mice, neither statistically significant nor in total (median) values. So reexpression of IRSp53 does not alter this parameter. The fact that there is no significant difference between wt-EGFP and cKO-IRSp53 mice is simply due to the large experimental variability for most of the data shown in this Figure. No conclusions can be drawn from these data.

Several other Figures, especially Fig. 3c, are plagued by this rather large experimental variability, and it appears difficult to draw any conclusions.

The neurons shown in Fig. 4b exhibit a very strong level of fluorescence, indicating strong overexpression. Compared to pictures shown in Fig. 1, where many cells expressing the transgene are shown, it is unclear why only very few cells are infected in Fig. 4b. The authors should explain this discrepancy. Were only cells with a very strong expression used for recordings?

Minor points.

Line 67 has a typo (flexed target gets?)

Materials and methods: SAN is not explained; it is not clear where on the optiprep gradient the AAVs were collected

Reviewer #3 (Remarks to the Author):

In this manuscript the authors have used an AAV penetrant to the BBB to drive adult IRSp53 re-expression in IRSp53-mutant mice. They found that the re-expression of IRSp53 in cKO improved nose-to nose interaction but not the total body contact time. Sociability was improved but not effects were observed on locomotor activity and anxiety behaviors.

The authors then decided to focus on layer 5pyramidal neurons within the PFC and report the effect of IRSp53 re-expression in adulthood on synaptic transmission.

Understand whether adult re-expression of certain genes in adulthood is sufficient to normalize brain function, is an important topic in psychiatry and the questions asked in this manuscript are relevant. Although the behavioral data are interesting and the analysis of glutamatergic transmission in the mPFC is pertinent and well justified, the conclusions are not supported by the data and the effect of adult re-expression on synaptic transmission are modest and mostly confusing. In particular, I am referring to the conclusion that re-expression of IRSp53 in adulthood partially rescue electrophysiological properties of layer 5 pyramidal neurons within the PFC and the conclusions regarding the presynaptic release properties at NMDA containing synapses. Furthermore, how do the authors explain that the over expression of IRSp53 in WT mice induces the same effects that the downregulation of the protein in cKO mice? The effects on the IPSCs are not well justified and the conclusion that IRSp53 overexpression in WT mice and IRSp53 re-expression in cKO mice induces similar decreases in transmission and similar increases in inhibitory transmission is confounding.

Furthermore, could the authors comment about the inability to test social novelty recognition in WT mice?

The link between the behavior and the electrophysiological changes in the mPFC is weak since the overexpression includes anterior cingulate cortex and hippocampus.

Point-by-point responses to reviewers' comments:

Reviewer #1 (Remarks to the Author):

This manuscript describes adult IRSp53 re-expression in IRSp53-mutant mice via a brain-blood barrier-penetrant AAV retro-orbital injection could rescue the social deficits and NMDAR-mediated excitatory synaptic transmission in the medial prefrontal cortex. Although the data are of high quality, I don't see any new conceptual /technical or mechanistical innovations. The authors found that re-expression IRSp53 in adult period restores IRSp53-dysfunction related phenotypes in mice, while no further mechanism was uncovered in this field. Since restored important genes at adult stages after the completion of brain development to normalize defective brain functions have been reported, such as MeCP2, GluN1, Shank3, SynGAP1, as mentioned in the introduction of this manuscript, there are not sufficient conceptual advances in this manuscript.

→ We agree with the reviewer's comments. However, we have to point out that there are also many genes in which adult gene restorations do not rescue major phenotypes, indicating that the outcome of adult gene re-expression depends on the genes and that each gene needs to be tested individually. This would be an important issue for potential gene therapy of each individuals in the future. In addition, our study attempts adult gene re-expression by a viral tool rather than Cre-mediated gene re-expression, trying to better mimic clinical situations. We clarified these points in the revised Introduction and Discussion.

Other issues:

1. What's the phenotype of adult deletion of IRSp53. Is it different from the IRSp53 knock-out from embryonic or early postnatal stages?

→ To address this question, we attempted a late IRSp53 deletion occurring during juvenile and adult stages by crossing IRSp53-flox mice with CaMKII-Cre mice. The results indicate that CaMKII-Cre-dependent IRSp53 deletion in mice leads to various behavioral deficits, including open-field hyperactivity, anxiety-like behaviors, and impaired social interaction (**Supplementary Fig. 2**), similar to Emx1-Cre-dependent IRSp53 deletion. Therefore, adult-stage IRSp53 expression seems to be required for normal brain function. We added related discussion points to the revised Discussion section.

2. The authors show the restored social deficits in IRSp53-mutant mice by IRSp53 re-expression. The mPFC has been strongly implicated in social regulation, so the authors chose the mPFC to detect NMDAR function, while I don't see obvious EGFP signals representing IRSp53 expression by BBB-penetrant PHP.eB in mPFC from Fig. 1c. The restored social deficits found in this manuscript may be attribute to other area.

→ We agree with the reviewer that IRSp53 re-expression in the mPFC region is not as strong as those observed in other cortical areas and the hippocampus. However,

IRSp53 signals can be detected in the mPFC brain region although at relative low levels; we made this clear in a revised figure (**Fig. 1f**). In addition, we agree with the reviewer that the behavioral rescue induced by adult IRSp53 re-expression is highly likely to involve multiple brain regions in addition to the mPFC because various brain regions have been implicated in social regulation; we clarified this in the revised Discussion section.

3. The data shown in Fig. 2e are inconsistent with the statistics in Fig. 2f. The author didn't observe obvious difference between a novel stranger and a familiar stranger, while the data in Fig. 2e show an obvious tendentiousness to S2.

→ We appreciate this careful comment and replaced the images in **Fig. 2e** with more representative ones.

4. Both deletion of IRSp53 and adult IRSp53 overexpression in WT mice induced an increase in the NMDAR/AMPA-EPSC ratio, this is a surprising result. It will be interesting to uncover the bidirectional regulation mechanism of IRSp53 in NMDAR functions.

→ We agree that these results (increased NMDA/AMPA ratio; **Fig. 4c-e**) are unexpected, although they are interesting and need additional clarification. We hypothesized in our previous paper that IRSp53 deletion in mice suppresses LTD- and NMDAR-dependent F-actin depolymerization in dendritic spines and also NMDAR-dependent long-term depression of NMDARs (not AMPARs) (Chung et al., Nat Neurosci, 2015). These changes would increase NMDAR function and the NMDAR/AMPA ratio in the mutant mice (shown as a diagram in **Supplementary Fig. 4**). We clarified this in Discussion.

IRSp53 overexpression in WT mice seems to induce excessive F-actin polymerization and generation of NMDAR-only silent synapses, as supported by our new results (**Fig. 4e,f**). This would similarly increase the NMDA/AMPA ratio (also shown in **Supplementary Fig. 4**). In contrast, IRSp53 re-expression in IRSp53-cKO mice did not affect silent synapses (**Fig. 4f,g**).

Reviewer #2 (Remarks to the Author):

The manuscript describes a „rescue experiment“ for a neurodevelopmental disorder. Mice lacking the gene coding for the insulin receptor substrate of 53 kDa (IRSp53) are treated with an AAV expression vector to correct expression of this gene, and are analysed on a behavioural as well as electrophysiological level.

IRSp53 deficient mice were originally described with a strong phenotype in learning and memory (as evident in the Morris water maze, and contextual fear conditioning assays). Later on it was also shown that these mice exhibit an autism related phenotype, exemplified by reduced social interactions. On a molecular/cellular level, these findings correlated with an enhanced NMDA receptor function, evident as increased NMDA/AMPA ratio. In humans, variants in the coding gene for this protein correlate in a few cases with neurodevelopmental disorders including autism and schizophrenia.

As these findings pointed to a neurodevelopmental disorder, it was rather interesting to see here whether reexpression of IRSp53 by means of a virus at the adult stage would lead to a proper rescue of the abovementioned phenotypic features. Importantly, conditional ko mice are used which lack IRSp53 expression only in excitatory/glutamatergic neurons, and the reexpression strategy makes sure that only these neurons are targeted by the transgene.

Initially the authors observe efficient expression of the protein, as evidenced by fluorescence microscopy. In behavioural assays, the reexpression of IRSp53 induces an impressive rescue of the social interaction, as evident in “nose-to-nose” contact of the mice (Fig. 2b). On the other hand, IRSp53 reexpression does not rescue the increased locomotor activity (open field assay), or the anxiety phenotype (elevated plus maze).

In electrophysiological experiments the authors claim that IRSp53 reexpression partially rescues the increased NMDA/AMPA ratio observed in ko mice. On the other hand, they also observed an increase in NMDA/AMPA ratio upon IRSp53 overexpression in wt mice.

Overall this is a very interesting study, as it addresses the important question whether and when it is possible to rescue the neurodevelopmental phenotype through reexpression of the target gene (which is mutated in the patient, or in this case in the mouse).

→ We appreciate the careful and encouraging summary of the reviewer.

Nevertheless, there are several problems associated with this manuscript which should be addressed.

Major points.

1. p. 5, bottom, and p.6 top, as well as Fig. 1a: the genetic approach is not well explained; I assume that the triangles in the vector map are LoxP sites; and that a recombination event is supposed to take place in cells where the genomic *Baiap2* is knocked out by Cre recombinase. But this is really not evident from the text.

→ To clarify the experimental procedures on *Emx1*-dependent embryonic IRSp53 deletion and PHP.eB-dependent adult IRSp53 re-expression, we replaced **Fig. 1a** with **Fig. 1a,c** to better visualize the Cre-dependent conditional knockout of IRSp53 in excitatory/glutamatergic neurons starting at embryonic stages (**Fig. 1a**) that is followed by Cre-dependent adult IRSp53 re-expression of IRSp53 by PHP.eB-IRSp53 injection (**Fig. 1c**). We clarified it the revised Results.

2. p. 6, and Fig. 1. There is no information regarding the 2A-peptide cleavage. As there appears to be almost perfect colocalization between HA-and GFP signals in Figure 1d, one might conclude that cleavage did not occur. Otherwise the HA-signal should be more distinct from the GFP signal, as HA-IRSp53 is supposed to be synaptically localized. A western blot analysis of affected brain regions would clarify this, and would also provide information regarding the levels of overexpression achieved compared to wt, and cKO animals.

→ We agree with the strong colocalization of the HA and GFP signals in the

hippocampus, probably because of their usually strong signals. However, enlargement of these signals in most of other brain regions such as the mPFC, shown in the new **Fig. 1f** (replacing Fig. 1d in the original manuscript), clearly indicates that HA and GFP distribution patterns are largely distinct, where GFP signals are mainly present in the cell bodies whereas HA signals, representing re-expressed IRSp53, are mainly present in neurites, as correctly pointed out by the reviewer. In addition, immunoblot analysis of the brain samples (cortex and hippocampus combined) from the mutant mice with adult IRSp53 re-expression indicate that the re-expressed IRSp53 band observed at ~55 kDa does not accompany a larger undigested band (~80 kDa; EGFP fused with HA-IRSp53) (**Fig. 1e** and **Supplementary Fig. 1a** [full-length immunoblots]). In addition, our quantification indicated that the levels of re-expressed IRSp53 proteins in the mutant mice were ~60% of WT levels (**Fig. 1e**).

3. Overall, there is no information regarding the efficiency of the viral approach. Which percentage of cells is infected, in particular which percentage of the glutamatergic/excitatory neurons?

→ To address this point, we performed double FISH experiments for EGFP and Vglut1/2 (markers of glutamate neurons) and found that ~64–85% of EGFP-positive neurons were Vglut1/2 positive (70.5% for prelimbic area, 64.2% for infralimbic area, 85.9% for primary motor area) and that ~55–77% of Vglut1/2-positive neurons were EGFP positive (55.3% for prelimbic area, 62.6% for Infralimbic area, 77.0% for primary motor area) (**Fig. 1g-i**).

4. Fig. 2c,d and Fig. 2e,f. These figures are not well explained. Assuming that the heat maps show representative behaviour of mice, one would assume that cKO-IRSp53 mice spend 90 % of their time with S1 and very little time with object O. The bar graph in 2d then gives a very different impression (a moderate 50 % increase in time spent with S1, compared to time with object O). Then the authors claim that in 2e, wt mice do not show a preference for stranger S2; but in fact the heat map shows very close interaction with S2 for the wt-EGFP positive control, very much in contrast to cKO-EGFP, which stays away from S2 as far as possible. Is the heat map only for a single mouse? Which is not representative for the whole cohort of mice?

→ We apologize for the inconvenience. We replaced the images in **Fig. 2c,e** with more representative ones.

5. Fig. 4. I do not think that, as claimed by the authors, there is a partial rescue of NMDA/AMPA ratio in Fig. 4d. There is no difference between cKO-EGFP and cKO-IRSp53 mice, neither statistically significant nor in total (median) values. So reexpression of IRSp53 does not alter this parameter. The fact that there is no significant difference between wt-EGFP and cKO-IRSp53 mice is simply due to the large experimental variability for most of the data shown in this Figure. No conclusions can be drawn from these data.

→ We fully agree. To address this point, we performed additional experiments to increase the n numbers for the NMDA/AMPA ratio data. Now the results clearly

indicate that adult IRSp53-reexpression in IRSp53-cKO neurons rescues the NMDA/AMPA ratio (**Fig. 4c,d**). We also changed the figure format from box-whisker plots to dot plots (with average + SEM; the same statistical results) to better visualize the distribution patterns of individual data points.

6. Several other Figures, especially Fig. 3c, are plagued by this rather large experimental variability, and it appears difficult to draw any conclusions. The neurons shown in Fig. 4b exhibit a very strong level of fluorescence, indicating strong overexpression. Compared to pictures shown in Fig. 1, where many cells expressing the transgene are shown, it is unclear why only very few cells are infected in Fig. 4b. The authors should explain this discrepancy. Were only cells with a very strong expression used for recordings?

→ Regarding the results in **Fig. 3c**, we agree with the reviewer. However and again, given that the box-whisker plots do not give precise information on the distribution of individual data points, we changed the bar graphs to dot plots (with average + SEM; the same statistical results) throughout the manuscript.

We enhanced the image in **Fig. 4b** in the original manuscript to emphasize that we made electrophysiological measurements in EGFP-positive neurons. In other words, these neurons are not as bright as they look; we clarified this point in the figure legend. In addition, the scarcity of neurons in the view field is not uncommon as you can see in our new images of EGFP-positive cells in the mPFC (**Fig. 1f**). The reviewer's impression that there are many EGFP-positive cells in the **original Fig. 1d** (now **Supplementary Fig. 1b**) may be associated with the hippocampal CA2 region where EGFP-positive cells were abundant, in agreement with the previous results (Hamodi et al., 2020), reflecting the nature of the PHP.eB AAV; we pointed out this in the figure legend.

Minor points.

1. Line 67 has a typo (flexed target gets?)

→ We appreciate; corrected.

2. Materials and methods: SAN is not explained; it is not clear where on the optiprep gradient the AAVs were collected

→ We explained HL-SAN in Methods. AAVs were collected in the Optiprep gradient of 40–60%; we clarified it in Methods.

Reviewer #3 (Remarks to the Author):

In this manuscript the authors have used an AAV penetrant to the BBB to drive adult IRSp53 re-expression in IRSp53-mutant mice. They found that the re-expression of IRSp53 in cKO improved nose-to nose interaction but not the total body contact time. Sociability was improved but not effects were observed on locomotor activity and anxiety behaviors.

The authors then decided to focus on layer 5pyramidal neurons within the PFC and

report the effect of IRSp53 re-expression in adulthood on synaptic transmission.

→ We appreciate the careful summary of the reviewer.

1. Understand whether adult re-expression of certain genes in adulthood is sufficient to normalize brain function, is an important topic in psychiatry and the questions asked in this manuscript are relevant. Although the behavioral data are interesting and the analysis of glutamatergic transmission in the mPFC is pertinent and well justified, the conclusions are not supported by the data and the effect of adult re-expression on synaptic transmission are modest and mostly confusing. In particular, I am referring to the conclusion that re-expression of IRSp53 in adulthood partially rescue electrophysiological properties of layer 5 pyramidal neurons within the PFC and the conclusions regarding the presynaptic release properties at NMDA containing synapses.

→ We fully agree. To address this point, we performed additional experiments to increase the n numbers for the NMDA/AMPA ratio data. Now the results clearly indicate that adult IRSp53-reexpression in IRSp53-cKO neurons rescues the NMDA/AMPA ratio (**Fig. 4c,d**). In addition, we deleted the figure panels (original **Fig. 4e-g**) in the original manuscript showing the results involving the analysis of co-efficient of variations because now the NMDA/AMPA ratio results are clear, and the results on co-efficient of variation could confuse readers.

2. Furthermore, how do the authors explain that the over expression of IRSp53 in WT mice induces the same effects that the downregulation of the protein in cKO mice?

→ We agree that these results (increased NMDA/AMPA ratio; **Fig. 4c-e**) are unexpected, although they are interesting and need additional clarification. We hypothesized in our previous paper that IRSp53 deletion in mice suppresses LTD- and NMDAR-dependent F-actin depolymerization in dendritic spines and also NMDAR-dependent long-term depression of NMDARs (not AMPARs) (Chung et al., 2015). These changes would increase NMDAR function and the NMDAR/AMPA ratio in the mutant mice (shown as a diagram in **Supplementary Fig. 4**). We clarified this in Discussion.

IRSp53 overexpression in WT mice seems to induce excessive F-actin polymerization and generation of NMDAR-only silent synapses, as supported by our new results (**Fig. 4e,f**). This would similarly increase the NMDA/AMPA ratio (also shown in **Supplementary Fig. 4**). In contrast, IRSp53 re-expression in IRSp53-cKO mice did not affect silent synapses (**Fig. 4f,g**).

3. The effects on the IPSCs are not well justified and the conclusion that IRSp53 overexpression in WT mice and IRSp53 re-expression in cKO mice induces similar decreases in transmission and similar increases in inhibitory transmission is confounding.

→ With regard to the similar decreases in mEPSC frequency in IRSp53-expressing WT mice and IRSp53-re-expressing IRSp53-cKO mice (**Fig. 5a**), our new results

indicate that IRSp53 overexpression in WT excitatory synapses increases the number of silent synapses (**Fig. 4e,f**). This seems to induce the translocation of some of the presynaptic release machinery from existing synapses to new silent synapses, and thus suppressing presynaptic release from existing synapses, as supported by the increased paired pulse facilitation ratios in IRSp53-overexpressing WT neurons (**Fig. 4g,h**) and hypothesized in a diagram (**Supplementary Fig. 4**).

Our previous study has reported a decrease in mEPSC frequency in IRSp53-cKO layer-5 pyramidal neurons up until week 12, likely including week 8 (Kim et al., 2020). Our data indicate that IRSp53 re-expression in IRSp53-cKO mice rescue the NMDA/AMPA ratio (**Fig. 4c,d**), but not the decreased mEPSC frequency. We clarified this in a new diagram (**Supplementary Fig. 4**).

With regard to the similar increases in inhibitory synaptic transmission in IRSp53-overexpressing WT mice and IRSp53-re-expressing IRSp53-cKO mice (**Fig. 5d-f**), related mechanisms remain unclear. However, IRSp53 is mainly expressed in glutamate (but not GABA) neurons in the cortex (Burette et al., 2014) and mainly localized at excitatory synapses (Abbott et al., 1999). In addition, the observed increase in mIPSCs occurs in the amplitude rather than the frequency. These results suggest that the change in mIPSC amplitude may involve cell-autonomous postsynaptic changes. We did not mention these points in Discussion for the lack of a reasonable amount of data.

4. Furthermore, could the authors comment about the inability to test social novelty recognition in WT mice?

→ We are not sure why WT mice were unable to differentiate a novel stranger and a familiar stranger (S2 vs. S1; **Fig. 2e,f**). On the other hand, the results from another three-chamber test performed on CaMKII-Cre;Irs53fl/fl mice indicated a normal recognition of a novel stranger (**Supplementary Fig. 2**), suggesting that the different results could stem from the distinct natures of different mouse lines (different cell-type-specific cKOs of the same gene in this case). We commented on this difference in Results.

The link between the behavior and the electrophysiological changes in the mPFC is weak since the overexpression includes anterior cingulate cortex and hippocampus.

→ We agree. We commented in Discussion that mPFC represents only one of many brain regions with adult IRSp53 re-expression.

References

- Abbott, M.A., Wells, D.G., and Fallon, J.R. (1999). The insulin receptor tyrosine kinase substrate p58/53 and the insulin receptor are components of CNS synapses. *J Neurosci* 19, 7300-7308.
- Burette, A.C., Park, H., and Weinberg, R.J. (2014). Postsynaptic distribution of IRSp53 in spiny excitatory and inhibitory neurons. *J Comp Neurol* 522, 2164-2178. 10.1002/cne.23526.
- Chung, W., Choi, S.Y., Lee, E., Park, H., Kang, J., Park, H., Choi, Y., Lee, D., Park, S.G., Kim, R., et al. (2015). Social deficits in IRSp53 mutant mice improved by NMDAR and mGluR5 suppression. *Nat Neurosci*. 10.1038/nn.3927.

Hamodi, A.S., Martinez Sabino, A., Fitzgerald, N.D., Moschou, D., and Crair, M.C. (2020). Transverse sinus injections drive robust whole-brain expression of transgenes. *Elife* 9. 10.7554/eLife.53639.

Kim, Y., Noh, Y.W., Kim, K., Yang, E., Kim, H., and Kim, E. (2020). IRSp53 Deletion in Glutamatergic and GABAergic Neurons and in Male and Female Mice Leads to Distinct Electrophysiological and Behavioral Phenotypes. *Front Cell Neurosci* 14, 23. 10.3389/fncel.2020.00023.

Reviewers' comments:

Reviewer #1 (Remarks to the Author):

After reviewing this revised manuscript and reading the authors responses, I think that the authors have addressed all of my comments satisfactorily.

Reviewer #2 (Remarks to the Author):

My main concerns regarding this manuscript have been met, so I have no further objections.

One minor thing which occurred to me while reading the title:

"Adult re-expression of BBB-penetrant IRSp53 rescues NMDA receptor function and social behavior in IRSp53-mutant mice"

IRSp53 is not BBB-penetrant. The virus is, I guess, but not the expressed protein. Maybe the authors want to be precise, and just delete the term "BBB-penetrant".

Reviewer #3 (Remarks to the Author):

The authors have addressed several points raised by the reviewers during the revision and the manuscript has improved.

Overall, I still find the result of electrophysiological results very confusing. The differences between 8 and 12 weeks and the increase ratio in WT mice when IRSp53 is overexpressed are not clearly justified or explained and the results of the experiments of silent synapses and paired pulse ratio are not convincing. The authors write "These results from IRSp53-re-expressing IRSp53-cKO neurons, together with the normal NMDA/AMPA ratio of naïve IRSp53-cKO mice at ~8 weeks and the increased NMDA/AMPA ratio of these neurons at ~12 weeks, suggest that IRSp53 re-expression in IRSp53-cKO neurons starting at ~8 weeks prevents the NMDA/AMPA ratio from being abnormally increased during 8–12 weeks, although it fails to prevent the presynaptic release to be abnormally increased. In addition, the results from IRSp53-overexpressing WT neurons suggest that IRSp53 overexpression induces increases in silent synapses and NMDA/AMPA ratios and a decrease in presynaptic release in non-silent/pre-existing synapses"

The conclusions are confusing, and the presented results are not supporting the conclusions. Furthermore, the discussion about the hypothetical sequence of the events is too speculative.

Regarding the figure2, the authors report that adult IRSp53 re-expression in IRSp53-cKO mice improved nose-to-nose interaction but not by total body contact time. At the end of the paragraph, the authors should write that These results collectively suggest that IRSp53 re-expression in adult IRSp53-cKO mice can improve some aspects of direct social interaction and social approach.

In figure 3, the elevated plus maze, does not reveal differences between groups. Therefore, the only indication of an anxiety phenotype is the result with the open field test and light dark task. This should be discussed.

Point-by-point responses to reviewers' comments:

Reviewer #1 (Remarks to the Author):

After reviewing this revised manuscript and reading the authors responses, I think that the authors have addressed all of my comments satisfactorily.

→ We appreciate the final comments of the reviewer.

Reviewer #2 (Remarks to the Author):

My main concerns regarding this manuscript have been met, so I have no further objections.

One minor thing which occurred to me while reading the title:

"Adult re-expression of BBB-penetrant IRSp53 rescues NMDA receptor function and social behavior in IRSp53-mutant mice"

IRSp53 is not BBB-penetrant. The virus is, I guess, but not the expressed protein. Maybe the authors want to be precise, and just delete the term "BBB-penetrant".

→ We appreciate the final comments of the reviewer. We agree that "BBB-penetrant" should be deleted from the title and accordingly changed the title of our manuscript from "Adult re-expression of BBB-penetrant IRSp53 rescues NMDA receptor function and social behavior in IRSp53-mutant mice" to "Adult re-expression of IRSp53 rescues NMDA receptor function and social behavior in IRSp53-mutant mice".

Reviewer #3 (Remarks to the Author):

The authors have addressed several points raised by the reviewers during the revision and the manuscript has improved.

→ We appreciate the final comments of the reviewer.

Overall, I still find the result of electrophysiological results very confusing. The differences between 8 and 12 weeks and the increase ratio in WT mice when IRSp53 is overexpressed are not clearly justified or explained and the results of the experiments of silent synapses and paired pulse ratio are not convincing. The authors write "These results from IRSp53-re-expressing IRSp53-cKO neurons, together with the normal NMDA/AMPA ratio of naïve IRSp53-cKO mice at ~8 weeks and the increased NMDA/AMPA ratio of these neurons at ~12 weeks, suggest that IRSp53 re-expression in IRSp53-cKO neurons starting at ~8 weeks prevents the NMDA/AMPA ratio from being abnormally increased during 8–12 weeks, although it fails to prevent the presynaptic release to be abnormally increased. In addition, the results from IRSp53-overexpressing WT neurons suggest that IRSp53

overexpression induces increases in silent synapses and NMDA/AMPA ratios and a decrease in presynaptic release in non-silent/pre-existing synapses". The conclusions are confusing, and the presented results are not supporting the conclusions. Furthermore, the discussion about the hypothetical sequence of the events is too speculative.

→ We fully agree with the reviewer that these conclusions in the Results section are confusing and speculative. We thus modified the conclusions as follows in the revised Results to carefully describe the experimental results; "These results collectively suggest that IRSp53-cKO neurons at ~12 weeks with IRSp53 re-expression started at ~8 weeks show decreased NMDA/AMPA and paired-pulse ratios and unaltered silent synapses, compared with control IRSp53-cKO neurons. In addition, WT neurons at ~12 weeks with IRSp53 overexpression started at ~8 weeks show increased NMDA/AMPA and paired-pulse ratios and increased silent synapses, compared control WT neurons."

In addition, we elaborated the hypothetical sequence of the events in the revised Discussion by adding the following sentences; "However, we should point out that these predicted events are highly speculative. One way to test this hypothesis is to perform an electron microscopic analysis of WT neurons over-expressing IRSp53 to see if the number of NMDAR-only silent synapses was indeed increased and the number of pre-existing synapses containing both NMDARs and AMPARs was decreased."

Regarding the figure2, the authors report that adult IRSp53 re-expression in IRSp53-cKO mice improved nose-to-nose interaction but not by total body contact time. At the end of the paragraph, the authors should write that These results collectively suggest that IRSp53 re-expression in adult IRSp53-cKO mice can improve some aspects of direct social interaction and social approach.

→ We agree with the reviewer that we need to be more precise and tone down the conclusion. We thus changed the text in the Results from "...can improve direct social interaction and social approach" to "...can improve some aspects of direct social interaction and social approach".

In figure 3, the elevated plus maze, does not reveal differences between groups. Therefore, the only indication of an anxiety phenotype is the result with the open field test and light dark task. This should be discussed.

→ We appreciate this comment. In response, we added the following discussion points to the revised Results. The positive baseline anxiety-like behaviors of IRSp53-cKO mice in the open-field and light-dark tests but not in the elevated plus-maze test suggest that different anxiogenic stimuli such as openness, height, and brightness may differentially induce anxiety-like behaviors, in line with the distinct natures of these tests (Belzung and Griebel, 2001; Ramos, 2008).

References:

Belzung, C., and Griebel, G. (2001). Measuring normal and pathological anxiety-like behaviour in mice: a review. *Behav Brain Res* 125, 141-149.

Ramos, A. (2008). Animal models of anxiety: do I need multiple tests? Trends Pharmacol Sci 29, 493-498. 10.1016/j.tips.2008.07.005.